# Mitochondrial VDAC1 Silencing in Urethane-Induced Lung Cancer Inhibits Tumor Growth and Alters Cancer Oncogenic Properties

**DOI:** 10.3390/cancers16172970

**Published:** 2024-08-26

**Authors:** Nataly Melnikov, Srinivas Pittala, Anna Shteinfer-Kuzmine, Varda Shoshan-Barmatz

**Affiliations:** 1Department of Life Sciences, Ben-Gurion University of the Negev, Beer Sheva 8410501, Israel; melnnata@post.bgu.ac.il (N.M.); srinivas.pittala@nih.gov (S.P.); 2National Institute for Biotechnology in the Negev, Ben-Gurion University of the Negev, Beer Sheva 8410501, Israel; shteinfe@post.bgu.ac.il

**Keywords:** apoptosis, cancer, cancer stem cells, lung cancer, Metabolism, Mitochondria, VDAC1

## Abstract

**Simple Summary:**

Cancer cells exhibit several key characteristics, including uncontrolled growth, altered metabolism, enhanced survival mechanisms, and resistance to apoptosis, all of which are crucial for their persistence. In our study, we explored the impact of disrupting the production of mitochondrial gatekeeper protein VDAC1 on lung cancer. We induced lung cancer in mice using the chemical urethane, which closely mimics human lung cancer in terms of genetic mutations and molecular changes. Using MRI to monitor the lung tumors, we found that inhibiting VDAC1 expression in this mouse model led to significant changes: reprogramming of cancer cell metabolism, reduced tumor growth, alterations in the tumor microenvironment, and elimination of cancer stem cells (CSCs). Additionally, treatment with a peptide derived from VDAC1 also inhibited tumor growth and decreased CSC markers. These findings suggest that targeting VDAC1, either through depletion or with a cell-penetrating peptide, could be a promising therapeutic approach for lung cancer.

**Abstract:**

Alterations in cellular metabolism are vital for cancer cell growth and motility. Here, we focused on metabolic reprogramming and changes in tumor hallmarks in lung cancer by silencing the expression of the mitochondrial gatekeeper VDAC1. To better mimic the clinical situation of lung cancer, we induced lung cancer in A/J mice using the carcinogen urethane and examined the effectiveness of si-m/hVDAC1-B encapsulated in PLGA-PEI nanoparticles. si-m/hVDAC1-B, given intravenously, induced metabolism reprogramming and inhibited tumor growth as monitored using MRI. Mice treated with non-targeted (NT) PLGA-PEI-si-NT showed many large size tumors in the lungs, while in PLGA-PEI-si-m/hVDAC-B-treated mice, lung tumor number and area were markedly decreased. Immunofluorescence staining showed decreased expression of VDAC1 and metabolism-related proteins and altered expression of cancer stem cell markers. Morphological analysis showed two types of tumors differing in their morphology; cell size and organization within the tumor. Based on specific markers, the two tumor types were identified as small cell (SCLC) and non-small cell (NSCLC) lung cancer. These two types of tumors were found only in control tumors, suggesting that PLGA-PEI-si-m/hVDAC1-B also targeted SCLC. Indeed, using a xenograft mouse model of human-derived SCLC H69 cells, si-m/hVDAC1-B inhibited tumor growth and reduced the expression of VDAC1 and energy- and metabolism-related enzymes, and of cancer stem cells in the established xenograft. Additionally, intravenous treatment of urethane-induced lung cancer mice with the VDAC1-based peptide, Retro-Tf-D-LP4, showed inhibition of tumor growth, and decreased expression levels of metabolism- and cancer stem cells-related proteins. Thus, silencing VDAC1 targeting both NSCLC and SCLC points to si-VDAC1 as a possible therapeutic tool to treat these lung cancer types. This is important as target NSCLC tumors undergo transformation to SCLC.

## 1. Introduction

Lung cancer is the second most common malignancy in both men and women, and accounts for 75–80% of cancer-related deaths, making it the leading cause of mortality worldwide [1]. It is believed, that both genetic, as well as family history, polymorphisms, and environmental risk factors are responsible for lung cancer, with smoking being the major risk factor for this disease [2,3]. In fact, approximately 87% of lung-cancer cases are caused by cigarette smoking due to its carcinogenetic properties [4].

Non-small cell lung cancer (NSCLC), a leading cause of cancer-related deaths, represents 85% and the small cell lung cancer (SCLC) 15% of all lung cancer cases. NSCLC comprises three main subtypes: large cell carcinoma (LCC), adenocarcinoma (AC) and squamous cell carcinoma (SCC) [5]. AC is localized in the lung periphery, while SCC is frequently localized centrally in the lung [6].

Tumors which lack the classic glandular or squamous morphology, but exhibit neuroendocrine differentiation are classified as SCLC. They are composed of small cell nuclear chromatin that is finely granular with scant cytoplasm and inconspicuous cell borders [7,8]. SCLC constitutes ~15–20% of all lung carcinomas and is considered the most aggressive subtype with a poor prognosis, due to its diagnosis only in advanced stages [9,10]. Furthermore, without treatment, SCLC causes fatality within 2–4 months [11].

SCLC can be classified as a neuroendocrine tumor (NET). NETs originate from endocrine cells, and are associated with the presence of secretory granules, biogenic amines, and polypeptide hormones. Their clinical behavior is extremely variable [12]. There is also a rare subtype of SCLC, termed combined SCLC (CSCLC) that is characterized by a combination of SCLC and NSCLC [13].

Lung cancer treatment includes surgery, radiation, and systemic anticancer drug therapies including chemotherapy, targeted drugs, and immunotherapy [14]. While immune checkpoint blockade has improved survival for many NSCLC patients, it has failed to obtain long-term benefits [15]. Although immune checkpoint inhibitors profoundly improve overall survival in NSCLC patients in stages I–IV [16], not all patients respond to immunotherapy, and others develop immunotherapy primary- and secondary-resistance; thus, treatment must be discontinued [16]. For these reasons, there is a strong need for new treatments.

Recently, we demonstrated that the voltage-dependent anion channel 1 (VDAC1) is highly expressed in different tumors, pointing to its significance in high energy-demanding cancer cells, and we developed several strategies targeting VDAC1 in cancers [17,18,19,20,21,22,23,24,25,26]. VDAC1 is a mitochondrial protein that controls cell energy, metabolic homeostasis, and apoptosis [17,18,19,20,21,22,23,24,25,26,27]. It mediates the transport of metabolites, ions, nucleotides, Ca^2+^, and more, thus, controlling the metabolic cross-talk between the mitochondria and the cytosol, regulating mitochondrial activity. VDAC1 also plays a key role in apoptosis, participating in the release of apoptotic factors from the mitochondria and regulating apoptosis by interacting with anti-apoptotic proteins [19,21,27].

Previously, we demonstrated that silencing VDAC1 expression using si-RNA reduced cellular ATP levels, cell proliferation of a panel of cell lines regardless of cell origin, and mutation status [18,19,20,22,23,28,29].

In established mouse models, silencing VDAC1 expression inhibited solid tumor development and growth in cervical, lung, bladder, and breast cancers, mesothelioma and glioblastoma tumors resulted in metabolic rewiring, leading to a reversal of their oncogenic properties, which inhibited tumor growth, invasivity, stemness, epithelial-to-mesenchymal transition (EMT), and angiogenesis [18,19,20,22,23,28]. The treated metabolism reprogrammed residual “tumor” showed decreased proliferation, an altered tumor-microenvironment [20]. Moreover, NGS and proteomics analyses demonstrated that over changes in the expression of thousands of genes, transcription factors, different proteins and epigenetic-related enzymes and factors [17,18,20,29,30]. These epigenetic modifications point to an interplay between metabolism and epigenetics and can explain the changes in the expression of thousands of genes, transcription factors, and proteins, and lead to cell differentiation toward less malignant lineages [17,18,19,20]. Thus, VDAC1 is a key regulator of metabolic and energy reprogramming that disrupts cancer energy and metabolism homeostasis by downregulating VDAC1, making it a potential anti-cancer therapy strategy [29].

Here, we used the urethane-induced tumors that closely resemble human lung cancers in terms of histopathology and progression and often exhibit genetic mutations and molecular alterations similar to those found in human lung cancer, such as mutations in the KRAS gene. We show that silencing VDAC1 expression in a urethane-induced lung cancer mouse model reprogrammed metabolism, inhibited tumor growth, modulated the tumor microenvironment, and eliminated cancer stem cells (CSCs). Similar results were obtained with SCLCs. Additionally, a VDAC1-based peptide inhibited tumor growth, and decreased expression of metabolism-related enzymes and of CSCs markers. The results suggest that VDAC1 depletion and its peptide can be a possible therapeutic strategy in the treatment of lung cancer.

## 2. Methods and Materials

### 2.1. Materials

The cell transfection reagents siLentFect and in-vivo JetPEI were obtained from Bio-Rad (Hercules, CA, USA) and from PolyPlus (Illkirch, France), respectively. 2′-O-methyl-modified m/hVDAC1-B-siRNA (si-m/hVDAC1-B) and non-targeted si-RNA (si-NT) were obtained from Genepharma (Suzhou, China). Matrigel matrix was purchased from Corning (Glendale, AZ, USA). Bovine serum albumin (BSA), poly (D, L-lactic-co-glycolic acid) (PLGA), polyethylenimine (PEI), polyvinyl alcohol (PVA), trypan blue, Triton X-100, Tween-20, hematoxylin, eosin, 4′,6-diamidino-2-phenylindole (DAPI), dimethyl sulfoxide (DMSO), and urethane were obtained from Sigma-Aldrich (St. Louis, MO, USA). Paraformaldehyde was purchased from Emsdiasum (Hatfield, PA, USA). Dulbecco’s modified Eagle’s medium (DMEM), Roswell Park Memorial Institute media (RPMI), and Dulbecco’s Phosphate Buffered Saline (DPBS) were obtained from Gibco (Grand Island, NY, USA). Normal goat serum (NGS), Hank’s Balanced Salt Solution (HBSS), Waymouth MB 752/1 medium, Diethyl pyrocarbonate (DEPC)-treated water, fetal bovine serum (FBS), trypsin, ethylenediaminetetraacetic acid (EDTA), a chemiluminescence detection kit for HRP (EZ-ECL), and the supplement penicillin–streptomycin were obtained from Biological Industries (Beit Haemek, Israel). Protease inhibitor cocktail was purchased from Millipore (Billerica, MA, USA). 3,3-diaminobenzidine (DAB) was obtained from ImmPact-DAB (Burlingame, CA, USA). Primary and secondary antibodies, their source, and dilutions are listed in Table 1.

Antibodies against the indicated protein, their catalogue number, source, and dilutions used in immunoblotting (WB), immunofluorescence (IF), and immunohistochemistry (IHC) experiments are presented below.

### 2.2. Cell Culture

A549 (epithelial cell, human lung non-small cell carcinoma), H69 (human lung small cell carcinoma), H446 (epithelial cells, human lung small cell carcinoma), and 2LL (mouse Lewis lung carcinoma) cell lines were obtained from American Type Culture Collection (ATCC) (Manassas, VA) and maintained according to ATCC instructions. The various cell culture media were supplemented with 10% FBS, 100 U/mL penicillin, and 100 μg/mL streptomycin. All cells were maintained in a humidified atmosphere at 37 °C and 5% CO_2_. Cell lines were routinely tested for mycoplasma contamination.

### 2.3. si-RNA Transfection

For si-RNA transfection, cells were seeded in 6-well plates (150,000–200,000 cells/well) to a 40–60% confluence and transfected with non-targeting si-RNA (si-NT) or 2′-O-methyl-modified si-m/hVDAC1-B using the siLentFect reagent, according to the manufactory instructions.

si-NT, **Sense:** 5′GCAAACAUCCCAGAGGUAU3′,

**Anti-sense:** 5′ AUACCUCUGGGAUGUUUGC3′.

si-m/hVDAC1-B, **Sense**: 5′GAAUAGCAGCCAAGUAUCAGtt3′

**Anti-sense**: 5′CUGAUACUUGGCUGCUAUUCtt 3′.

The underlined nucleotides are 2′-O-methyl-modified.

### 2.4. Preparation of si-RNA Loaded–PLGA-PEI Nanoparticles

The PLGA-PEI-PVA-siRNA-loaded nanoparticle complexes were prepared using the solvent displacement method [31] with some modifications.

To the polyethylenimine (PEI)-aqueous solution (10 mg/mL in nuclease-free water), PVA solution was added (1 mg/mL) and mixed well by vortexing. si-NT or si-m/hVDAC1-B solutions in nuclease-free water (50 µM) added to the PEI-PVA solution and mixed well by vortexing. Samples were incubated in a 37 °C water bath for 30 min. After incubation, the PEI-PVA-siRNA mixture was added to a PLGA-acetone solution (10 mg/mL) by pipetting. Subsequently, primary emulsion, PLGA-PEI-siRNA, was added in a drop-wise manner (0.5 mL/min) to a glass beaker containing nuclease-free water with PVA solution. The mixture was continuously stirred with a magnetic stirrer at room temperature (RT) in a chemical hood for 2–3 h until complete evaporation of the organic solvent (no acetone smell) occurred. The milky nanoparticle mixture was centrifuged at 20,000× *g* at 4 °C for 30 min, and the pellet was re-suspended in nuclease-free water and centrifuged at 20,000× *g* at 4 °C for 30 min. The resulting nanoparticles, PLGA-PEI si-m/hVDAC1-B, were re-suspended in nuclease-free PBS, and the suspension was aliquoted and centrifuged, and the pellet was stored at −20 °C until further use.

On the treatment day, the pellet was first incubated for 30 min at RT, and then was re-suspended in PBS in nuclease-free water, vortexed, and incubated at 37 °C for 10 min.

Mice were treated by intravenous (i.v) injection of 100 μL of the nanoparticles containing si-NT or si-m/hVDAC1-B suspension in order to reach the desired concentration in the blood.

### 2.5. Peptide Solution Preparation

VDAC1-based peptide, Retro-Tf-D-LP4 (>95% purity), is composed of a sequence derived from VDAC1 with amino acids (VDAC1 sequence between 216–219 amino acids) added to the N- and C-terminus to form a tryptophan zipper thus to a loop-like shape (sequence is in italic). This peptide was fused to the cell-penetrating sequence (Tf, sequence is underlined) that is recognized by transferring receptor [32] that mediates its transport into the cell. Amino acids in the D configuration are shown in bold:

***KWTWK***-216-**NSNGATWALNVATE LKK**-199-***EWTWS***HRPYIAH (34 residues)

The peptide was first dissolved in 100% DMSO, and then diluted with sterilized water to a final concentration of 25 mg/mL in 5% DMSO. Peptide concentrations were determined using absorbance at 280 nm and the specific molar excitation coefficient, as calculated based on amino acid composition. The prepared peptide was stored in aliquots at −20 °C.

For the experiment, the final concentrations of DMSO in the blood of control and peptide-treated mice were less than 0.5%.

### 2.6. Urethane-Induced Lung Tumor Mice Model

Mice used for experiments were sex, weight, and age matched. Six-week-old A/J strain mice (weighing 20–22 g) were obtained from Envigo (Beer Sheva, Israel). Urethane (ethylcarbamate) was used to induce lung tumors in the mice, according to published protocol [33]. Urethane was dissolved in PBS at 100 mg/mL, and aliquots were stored at −20 °C.

Following one week of acclimatization, the mice were injected intraperitoneally (i.p.) with 1 mg of urethane per gram of body weight in a total volume of 200 µL of saline. The experiment protocol was planned according to tumor development after the urethane injection [34] (Figures 2A, 3A and 9A).

The tumor development was monitored with magnetic resonance imaging (MRI), was carried out three times during the experimental time, as described below, and tumor volume was calculated using Vivo Quant software (version 2.5patch2). When the average tumor volume reached 1.5–2.3 mm^3^, mice were randomized into four matching groups. The mice were i.v injected 3 times a week with PLGA-PEI-si-m/hVDAC1-B (8 mice) starting at week 19, 21 or week 23, or PLGA-PEI-si-NT (200 or 350 nM, 7 mice). Retro-Tf-D-LP4 peptide was i.v injected to a final concentration of 18 mg/Kg (0.4 mg/mouse/100 μL), diluted in HBSS containing 5% DMSO (n = 4 or 5). The blood final concentration of DMSO in control and peptide-treated mice was 0.3%. The mice were injected three times a week.

At the experiment end point, the mice were sacrificed using CO_2_, and their lungs were excised, fixed and processed for immunohistochemistry (IHC) or immunofluorescence (IF). Two independent experiments with different PLGA-PEI-si-m/hVDAC1-B concentrations were performed.

Experimental protocols were approved by the Institutional Animal Care and Use Committee.

### 2.7. MRI Tumor Monitoring

Tumor development in the lungs of the mice was tracked with MRI using an M7 1-Tesla compact ICON system (Aspect Imaging, M7, Beer Sheva, Israel), equipped with a set of 80-mm application-specific radiofrequency (RF) mouse body coils. For in-vivo imaging, animals were maintained in an anesthetized state with 1.5% isoflurane in O_2_. Each mouse was placed in an animal handling system with heating, where physiological signals, such as breath rate, were monitored throughout the experiment to ensure animal well-being. The system parameters included a fast spin echo with a repetition time of 6,624,432 ms and an echo time of 55.05 ms. Eighteen coronal slices of 1 mm with a gap of 0.1 mm, a field of view (FOV) of 90 × 30 mm, and an acquisition time of 12.38 min were collected.

### 2.8. Xenograft Mouse Model

Athymic (C57BL/6JOlaHsd) 6-week-old male mice (weight ~20–25 g) were obtained from Envigo (Israel), and given a week to acclimate. H69 cells (2 × 10^6^/mouse) in a serum-free media were mixed with matrigel solution 1:1 (*v*/*v*), and the mixed solution was injected subcutaneously (s.c) into the hind leg flank of each mouse. Mouse weight was monitored twice a week for a period of 41 days. Twenty-one days post-cell inoculation, the developed tumors were measured in two dimensions with a digital caliper, and the tumor volume was calculated as follows: tumor volume = (X^2^ × Y)/2, where X and Y are the short and long tumor dimensions, respectively. The mice with xenografts reaching a volume of 50–80 mm^3^ were divided into two groups with similar average tumor volume (7 mice/group); one group was treated with si-NT, and the second group with si-m/hVDAC1-B. Each treatment substance was injected intratumorally every 3 days with siRNA mixed with the in-vivo JetPEI reagent, according to the supplier instructors (200 nM final concentration, 2 boluses).

At the end point of the experiment, the mice were sacrificed using CO_2_, and the tumors were excised, fixed, and processed for IHC analysis.

Experimental protocols were approved by the Institutional Animal Care and Use Committee.

### 2.9. Immunohistochemistry and Immunofluorescence of Tumor Tissue Sections

IHC and IF staining were performed on 5-μm-thick formalin-fixed and paraffin-embedded tumor tissue sections. The sections were deparaffinized by incubation at 60 °C for 1 h. Thereafter, the tissue sections were rehydrated with a graded ethanol series (100–50%). Antigen retrieval was performed in 0.01 M citrate buffer (pH 6.0). After washing sections in PBS containing 0.1% Triton-X100 (pH 7.4, PBST), non-specific antibody binding was reduced by incubating the sections in a blocking buffer (10% NGS, 1% BSA, and 0.1% Triton) for 2 h. After decanting excess serum, sections were incubated overnight at 4 °C with primary antibodies (sources and dilutions used detailed in Table 1) in an antibody buffer (5% NGS, 1% BSA).

For the IHC analysis, endogenous peroxidase activity was blocked by incubating the sections in 3% H_2_O_2_ for 15 min. After washing thoroughly with PBST buffer, the sections were incubated for 2 h with HRP-conjugated secondary antibodies. Sections were washed five times in PBST, and the peroxidase reaction was subsequently visualized by incubation with 3,3-diaminobenzidine (DAB).

After rinsing in water, the sections were counterstained with hematoxylin, and mounted with EUKITT mounting medium (Orsatech, London, UK). Finally, the sections were scanned with a Pannoramic MIDI II microscope slide scanner (3DHISTECH Ltd., Budapest, Hungary), and images were collected at the same magnification and light intensity. Non-specific control experiments were carried out using the same protocols but omitting incubation with the primary antibodies. Hematoxylin and-eosin (H&E) staining was performed as described in [35].

For the IF analysis, after washing with PBST, sections were incubated with fluorescent-tagged secondary antibodies (Table 1) for 2 h at RT in the dark. Following a wash with PBST, sections were incubated with DAPI (0.07 μg/mL) for 15 min in the dark, washed, mounted with fluoroshield mounting medium (Im-munobioscience, Mukilteo, WA, USA), and imaged by confocal microscopy (Olympus 1X81) or with a Pannoramic MIDI II scanner. To represent the whole tumor area, images were taken from different fields of the tumor section.

A quantification analysis of IHC-stained slides was done using Case Viewer and of the IF slides using ImageJ software (version 1.54j).

### 2.10. Protein Extraction, Gel Electrophoresis, and Immunoblotting

Cells were lysed in lysis buffer (50 mM Tris- (version 16.0.9.0)HCl, pH 7.5, 150 mM NaCl, 1 mM EDTA, 1.5 mM MgCl_2_, 10% glycerol, 1% Triton-X100) supplemented with a protease inhibitor cocktail (Calbiochem, San Diego, CA, USA) and incubated on ice (30 min). Then cell lysate was centrifuged (10 min, 15,000× *g* at 4 °C), and the supernatant was used for immunoblotting.

Protein samples (10–20 μg) were resolved by SDS polyacrylamide gel electrophoresis and immunoblotted using the selected primary antibodies (Table 1), followed by incubation with horseradish peroxidase (HRP)-conjugated secondary antibodies. HRP activity was determined using a chemiluminescence detection kit. Band intensity was quantified using FUSION-FX (Vilber Lourmat, France) software (version 16.0.9.0).

### 2.11. Statistical Analysis

The data are shown as the mean ± SEM of at least three independent experiments unless specified differently. Significance of differences was calculated by a two-tailed Student’s *t*-test using the T-Test function provided by Microsoft Excel. Statistical significance is reported at *p* ≤ 0.05 (*), *p* ≤ 0.01 (**), or *p* ≤ 0.001 (***). Tumor volume data were analyzed using a one-way repeated measures ANOVA.

## 3. Results

Previously, we showed that silencing VDAC1 expression using human-specific siRNA (si-hVDAC1) resulted in the inhibition of cancer cell growth, both in vitro and in vivo in mouse intracranial and xenograft models of human glioblastoma, lung cancer, triple negative breast cancer, mesothelioma, and bladder cancers [17,18,19,20,22,23,28]. The results of these studies showed that tumor metabolic rewiring results in a reversal of their oncogenic properties, leading to inhibited tumor growth, invasivity, stemness, EMT, and angiogenesis [17,18,19,20,22,23,28,29].

In this study, we utilized a chemically-induced lung cancer using the carcinogen urethane that produces tumors that are similar to human clinical situation of lung cancer in terms of histology and molecular characteristics, along with si-m/hVDAC1-B, which recognizes both mouse and human VDAC1. We demonstrated that the inhibition of VDAC1 expression led to reprogramming cancer cell metabolism, markedly inhibited tumor growth of both NSCLC and SCLC, and eliminated CSCs associated with tumor recurrence and metastasis. Additionally, we demonstrated that the VDAC1-based peptide inhibited tumor development in urethane-induced lung tumors.

### 3.1. VDAC1 Overexpression in Human Lung Cancer Tissue and Silencing Its Expression by si-m/hVDAC1-B in Human and Mouse Lung Cancer Cell Lines

The expression levels of VDAC1 in the lung cancer tissue array comprised 10 samples from healthy human donors and 31 from lung cancer patients that were analyzed by IHC using anti-VDAC1 antibodies. Representative images from healthy and lung cancer tissues were sub-grouped according to staining intensity, with 15% of the cases showing low, 40% medium, and 45% high VDAC1 expression levels (Figure 1A). The results clearly show that lung cancer tissues express high VDAC1 levels compared to those in healthy lung tissues (Figure 1A).

The silencing of VDAC1 expression in human lung cancer A549 and H446 cell lines and in the mouse lung cancer 2LL cell line was conducted using a modified siRNA, si-m/hVDAC1-B, that recognizes both human and mouse VDAC1. VDAC1 expression levels were analyzed at protein levels following immunoblotting and quantitative analysis (Figure 1B–E). si-m/hVDAC1-B markedly decreased VDAC1 expression levels in the A549 and H446 cell lines by 90–95% (50 nM or 75 nM), and in the mouse lung cancer 2LL cell line by 60% (50 nM or 75 nM) (Figure 1B–E).

These results indicate that VDAC1 can be silenced using the newly developed siRNA, si-m/hVDAC1-B.

### 3.2. PLGA-PEI-si-m/hVDAC1-B Inhibits Urethane-Induced Lung Cancer in A/J Mice

To study the effect of si-m/hVDAC1-B on a lung cancer mouse model that better mimics the clinical situation of lung cancer, we induced lung cancer in A/J mice using the carcinogen urethane and monitored tumor growth by MRI.

The main limitation of the siRNA therapeutic features is their rapid degradation in plasma and cellular cytoplasm, resulting in a short half-life. Moreover, siRNA molecules cannot effectively penetrate into the cell [36]. In view of this, the use of a carrier system for its delivery is important. Thus, for treating lung tumors, we encapsulated si-NT or si-m/hVDAC1-B into PLGA-PEI nanoparticles. Two independent experiments of urethane-induced lung cancer were performed with a calculated final blood concentration of either 200 nM or 350 nM PLGA-PEI-si-m/hVDAC1-B.

The protocol used (Figure 2A) indicates that at about 18 weeks post-urethane injection, mice start to develop MRI-visible tumors, with clear tumors in the lungs shown at week 19 (Figure 2B).

Subsequently, i.v treatment with 200 nM of PLGA-PEI-si-m/hVDAC1-B or PLGA-PEI-si-NT was begun. Following 18 weeks of treatment, three times a week, mice were sacrificed, their lungs were removed and fixed, and sections were subjected to H&E staining or IHC. The results show that lungs treated with PLGA-PEI-si-NT contained many large-sized tumors, while in the PLGA-PEI-si-m/hVDAC-B-treated group, the tumor number markedly decreased (Figure 2C). Analysis of the tumor area and number of tumors showed an 86% and 63% decrease, respectively, following PLGA-PEI-si-m/hVDAC1-B treatment (Figure 2D,E). Moreover, the remaining “tumors” in the PLGA-PEI-si-m/hVDAC1-B group were localized at the lung surface borders, possibly suggesting that less PLGA-PEI-si-m/hVDAC1-B reached the lung surface.

The morphology of the tumor and non-tumor areas in the mouse lungs differed (Figure 2F). A tumor area showed morphological atypia [37], while histologic sections of a non-tumor area demonstrated an alveolar architecture [38].

In a second experiment, i.v. treatment with a 350-nM blood concentration of PLGA-PEI-si-m/hVDAC1-B or PLGA-PEI-si-NT was performed according to a modified protocol, where the treatment was started at two time-points, at week 21 (group A) and at week 23 (group B) post-urethane treatment (Figure 3A), and tumor development was monitored using MRI. MRI images of the lungs from representative mice showed tumors as white spots that were highly visible in the PLGA-PEI-si-NT-treated group and much less visible in images of the lungs of the PLGA-PEI-si-m/hVDAC1-B-treated mice (Figure 3B).

Using the appropriate MRI program, tumor volumes in the lungs were calculated for the two groups and showed a marked decrease in the tumor volume in the two PLGA-PEI-si-m/hVDAC1-B treated mice groups (Figure 3C). In group A, at week 30, the tumor volume was inhibited by 73%, and at week 33, the tumors had grown over 2-fold, whereas the tumor volume had decreased by 86% in the PLGA-PEI-si-m/hVDAC1-treated mice.

In group B, where the treatment with PLGA-PEI-si-NT or PLGA-PEI-si-m/hVDAC1-B was started two weeks later relative to group A, the inhibition of the tumors was 69%. IF staining of paraffin-embedded, formaldehyde-fixed lung sections showed significant silencing of VDAC1 expression in the PLGA-PEI-si-m/hVDAC1-B relative to its level in the lung tumors of the PLGA-PEI-si-NT-treated mice (Figure 3D).

The expression levels of the cell proliferation factor, KI-67, markedly decreased (73%), as shown by IHC staining and quantitative analysis (Figure 3E,F).

Taken together, these results indicate that PLGA-PEI-si-m/hVDAC1-B given by i.v. is delivered to the lungs, and decreases VDAC1 expression, inhibiting cell proliferation and tumor growth.

### 3.3. PLGA-PEI-si-m/hVDAC1-B Altered the Expression of Metabolism-Related Proteins in Tumors of Urethane-Induced Lung Cancer

Next, the expression levels of several metabolism-related proteins were analyzed in fixed lung sections of the PLGA-PEI-si-m/hVDAC1-B- and PLGA-PEI-si-NT-treated mice using IHC and specific antibodies (Figure 4). The expression levels of glucose transporter (Glut 1) in lung sections from the PLGA-PEI-si-NT-treated mice showed high levels of Glut 1 in the tumors, while in residual tumors in the lungs of the PLGA-PEI-si-m/hVDAC1-B-treated mice, these levels were low (Figure 4A). These results were further demonstrated in images that were magnified 250-fold and in the quantitative analysis (Figure 4A,B).

The expression levels of the glycolytic enzyme hexokinase 1 (HK-I) and the mitochondria protein VDAC1 were analyzed using specific antibodies (Figure 4C–F). The levels of these proteins were highly reduced in the PLGA-PEI-si-m/hVDAC1-B mice, relative to their levels in the PLGA-PEI-si-NT-treated mice (Figure 4C–F). Quantitative analysis of IHC staining intensity showed that the Glut 1, HK-I, and VDAC1 staining intensity in the PLGA-PEI-si-m/hVDAC1-B mice decreased by 60–68% relative to their levels in the PLGA-PEI-si-NT-treated mice (Figure 4).

The results are consistent with alterations in cell metabolism that occur with decreased uptake of glucose, glycolysis, and VDAC controlling mitochondrial activities, in agreement with the concept that cancer cells use a combination of glycolysis and mitochondrial activities to produce energy [39,40].

### 3.4. PLGA-PEI-si-m/hVDAC1-B Treatment of Urethane-Induced Lung Cancer Altered the Expression of CSC Markers

Mice lung tissues were analyzed for the expression levels of CSC markers using IHC and specific antibodies. Lung CSC markers include the transcription factor SOX2 [41], the protein aldehyde dehydrogenase isoform 1 (ALDH1) [42], and the cell surface markers CD133 and CD44 [43,44].

The expression levels of SOX2 in tumor and tumor-free tissue in the lungs of PLGA-PEI-si-m/hVDAC1-B- and PLGA-PEI-si-NT-treated mice using specific antibodies showed that SOX2 levels were higher in the tumor-free than in the tumor lung tissue in both the PLGA-PEI-si-m/hVDAC1-B- and PLGA-PEI-si-NT-treated mice (Figure 5A).

Similarly, the staining of lung sections from PLGA-PEI-si-m/hVDAC1-B- and PLGA-PEI-si-NT-treated mice with antibodies specific to ALDH1 member A1 (ALDH1A1) showed that in the tumor-free areas, the expression levels of ALDH1A1 were higher relative to its levels in the tumor areas, which showed no significant staining (Figure 5B).

These findings suggest that ALDH1A1 and SOX2 are not expressed in tumors produced in the lung due to urethane treatment. This option is further discussed in the Discussion section.

The levels of CD133 and CD44 as revealed with specific antibodies, were reduced in lung sections from PLGA-PEI-si-m/hVDAC1-B-treated mice, relative to their levels in PLGA-PEI-si-NT-treated mice (Figure 5C,D). Quantification of IHC staining intensity showed about a 60% and 38% decrease in CD133 and CD44, respectively, relative to their staining levels in PLGA-PEI-si-NT-treated mice (Figure 5D). Thus, PLGA-PEI-si-m/hVDAC1-B treatment of mice reduced CSCs.

### 3.5. Urethane Treatment Induces NSCLC and also SCLC Cancers, and both Were Inhibited by PLGA-PEI-si-m/hVDAC1-B

Analysis of the H&E staining of lung sections of A/J mice treated with urethane and PLGA-PEI-si-NT showed two types of tumors that differed in morphology of the stained tumors in the lung, with different cell sizes and organization within the tumor (Figure 6A(a,b)). These different tumors may represent SCLC and NSCLC, which we termed “a” and “b”. Moreover, it might present a combination of SCLC and NSCLC components, defined as combined small cell lung carcinomas (CSCLCs) [13]. The “classic” morphology of SCLC includes small, round, ovoid, and spindle-shaped cells with finely granular nuclear chromatin, scant cytoplasm, and ill-defined cell borders [45,46].

In order to verify whether these morphologically different tumors represent SCLC and NSCLC, we performed IHC staining of lung tissue sections from PLGA-PEI-si-NT-treated mice for markers proposed to distinguish between SCLC and NSCLC (Figure 6B–D). The markers of SCLC, including KI-67 and synaptophysin [10,11,13], were IHC stained using antibodies specific to these proteins (Figure 6B,C). The IHC stained tumors showed two areas within the same lung tumor with different staining intensities, with their levels were increased over 2.5- fold in one area relative to the other (Figure 5D), suggesting that these differently stained areas in the tumor represent CSCLC.

As PLGA-PEI-si-m/hVDAC1-B inhibited the growth of all urethane-induced tumors, we concluded that it inhibited the growth of both SCLC and NSCLC. Thus, we evaluated the effects of PLGA-PEI-si-m/hVDAC1-B on an SCLC xenograft.

### 3.6. si-m/hVDAC1-B Inhibited Tumor Growth in an SCLC Xenograft Mice Model

With a view to evaluate the effects of silencing VDAC1 expression using si-m/hVDAC1-B on SCLC-derived tumors, we established an SCLC xenograft mouse model using the H69 cell line (human SCLC) in mice (Figure 7C). si-m/hVDAC1-B treatment of these cells resulted in inhibition of VDAC1 expression (Figure 7A,B). Cells were injected into the mice, and when the tumor volume reached 50–80 mm^3^ (day 21), the mice were divided into two groups and intratumorally injected every three days with si-NT or si-m/hVDAC1-B mixed with in-vivo Jet-PEI reagent (Figure 7C,D). Already, after the first week of treatment, a difference in the tumor volumes between the two groups was observed. In the si-NT group, the average volume was approximately 123 mm^3^, while in the si-m/hVDAC1-B group, it was 77 mm^3^, i.e., an inhibition of about 37%. Subsequently, after the final injection, at 41 days post-inoculation, in the si-NT group, the average volume increased to 968 mm^3^, while in the si-m/hVDAC1-B group, the tumor volume was 248 mm^3^, i.e., an inhibition of 74%. Thus, si-m/hVDAC1-B treatment significantly inhibited SCLC tumor growth. A photograph of the tumors after sacrificing is shown in Figure 7D.

Next, VDAC1 expression levels in the tumors were analyzed using IHC staining (Figure 7E,F). The si-m/hVDAC1-B-stained section showed a decrease of about 70% in the VDAC1-staining intensity in comparison to that of the si-NT-treated tumors.

In order to verify the effect of VDAC1 on cell proliferation, tumor sections were stained with specific antibodies against the cell proliferation marker KI-67 (Figure 7G,H). The si-m/hVDAC1-B-stained tumor sections showed a decrease in KI-67 positive cells of about 75% in comparison to their levels in the si-NT-treated tumors.

The expression levels of metabolism-related enzymes Glut 1, the TCA-cycle enzyme citrate synthase (CS), and ATP synthase subunit 5a (ATP syn-5a) were analyzed by IHC staining the tumors treated with si-NT or with si-m/hVDAC1-B using specific antibodies (Figure 8A,B), The levels of these were reduced in the si-m/hVDAC1-B-treated tumors by 75%, 60%, and 50%, respectively, relative to the levels in the si-NT-treated tumors (Figure 8A,B). The results agree with alterations in the tumor metabolism and with the concept that cancer cells use not only glycolysis, but also mitochondria to produce energy [39,40].

As our previous study showed that VDAC1-tumor depletion eliminated CSCs [18,47], the tumors treated with si-NT or with si-m/hVDAC1-B were analyzed for the expression levels of CSC markers CD133 and CD44 in SCLC [48]. The expression levels of CD133 and CD44 were reduced in the si-m/hVDAC1-B-treated tumors relative to the levels in the si-NT-treated tumors (Figure 8C). Quantification of the protein levels showed about a 66% decrease of both CD44 and CD133 relative to their levels in the si-NT-treated tumors (Figure 8D).

### 3.7. VDAC-1-Based Peptide, Retro-Tf-D-LP4, Inhibited Tumor Growth in Urethane-Exposed Mice

Previously, we showed that VDAC1-drived peptides effectively induced cell death in vitro and in vivo [24,25,26]. The Retro-Tf-D-LP4 peptide inhibited tumor growth, induced apoptosis, reduced metabolic enzyme expression, and increased pro-apoptotic protein expression [24,25,26]. The effects of this peptide on NSCLC and SCLC, as induced by urethane in a mouse model, were tested (Figure 9).

Urethane-treated mice were i.v.-treated with the Retro-Tf-D-LP4 peptide (18 mg/Kg) or with DMSO as a control (see Methods section) (Figure 9A). Following tumor visualization by MRI at week 15, peptide treatment was begun at week 20 by i.v. injection three times a week. Following two weeks of treatment (week 25), MRI imaging showed a difference in tumor volume in the lungs. In the control group, total tumor volume per mouse was approximately 2.3 mm^3^, while in the Retro-Tf-D-LP4 peptide-treated group, tumor volume per mouse was 0.56 mm^3^, i.e., a reduction in the tumor volume of about 76%. Subsequently, after the last injection at week 30, in the control group, the total volume per mouse was about 25.5 mm^3^, while in the Retro-Tf-D-LP4-treated group, it was only 10.8 mm^3^, i.e., an inhibition of 58% (Figure 9B). Thus, Retro-Tf-D-LP4 peptide injected intravenously reached the lungs and inhibited tumor growth.

Next, VDAC1 and CS expression levels were analyzed using IF staining of sections from lungs of urethane-treated mice treated with the Retro-Tf-D-LP4 peptide. This analysis showed a decrease of about 68% and 70% in the expression levels of VDAC1 and CS, respectively, relative to their levels in the lungs of the urethane-treated control mice (Figure 9C–E). In addition, the expression levels of CSC marker CD44 were reduced in lung tumor sections from the Retro-Tf-D-LP4-treated group in comparison to its expression in sections from the control mice (Figure 9F,G).

## 4. Discussion

It is well-established that cancer cells reprogram their metabolism to provide the energy needed to promote tumor malignancy [49]. As such, targeting metabolic pathways presents a promising but complex strategy due to the inherent metabolic plasticity, redundancy, and adaptability of cancer cells. However, focusing on VDAC1—an essential protein that integrates cell metabolism with mitochondrial and cellular energy functions, and coordinates mitochondrial activities with other cellular processes [27,50], appears to address these challenges.

In previous studies across various cancer mouse models, including lung, bladder, breast cancers, glioblastoma, and mesothelioma, silencing VDAC1 expression reversed oncogenic properties, including reprogrammed metabolism, inhibited angiogenesis, epithelial-to-mesenchymal transition (EMT), invasiveness, and stemness [17,18,19,20,21,22,23,28,29,30,51]. Additionally, VDAC1 depletion led to changes in the expression of transcription factors (TFs), metabolic processes and the epigenetic landscape, affecting signaling pathways related to cancer hallmarks [17,18,20,29,30].

We have demonstrated that both siRNA against VDAC1 and VDAC1-based peptides are effective in inhibiting tumor growth in several cancer models, were the si-hVDAC1 encapsulated in PLGA-PEI nanoparticles, crossed the blood–brain barrier (BBB) and reached tumors in the brain [17,18,19,25,26,47]. Here, we used urethane-induced lung cancer in a mouse model, representing high similarity with human lung carcinogenesis [52], to demonstrate that siRNA recognizing both human and murine VDAC1, si-m/hVDAC1-B, when encapsulated into PLGA-PEI nanoparticles and given intravenously, reached the lung-silenced VDAC1 expression and led to inhibition of tumor growth (Figure 2 and Figure 3).

### 4.1. PLGA-PEI-si-m/hVDAC1-B as a Potential Treatment for Lung Cancer

Tumorigenesis requires alterations in cellular metabolism and bioenergetics of the cancer cells. VDAC1 as a mitochondria-gate keeper regulating the flux of metabolites and ions between the mitochondria and the cytoplasm, regulates cancer cell growth. Depletion of VDAC1 using si-hVDAC1 affects cancer cell metabolism, leading to inhibited cell proliferation in vitro and in vivo. In previous studies, we showed that silencing human VDAC1 in an NSCLC xenograft mouse model inhibited tumor growth and altered oncogenic properties [18,19,20].

In this study, we focused on the effects of metabolic reprogramming via silencing VDAC1 expression in a carcinogenic urethane-induced lung cancer model. This model closely mimics human lung cancer as it develops over a long period of time via mutations induced by urethane. Approximately 80% of the tumors induced by urethane treatment contain Kras^Q61L^ mutations and other genetic and epigenetic events affecting tumor progression [33], such as the mutation of p53 [53], mislocalization of p27 [54], altered DNA methylation [55], and loss of p19Arf expression [56]. Importantly, mutations like KRAS and p53 have been observed in many cases of human lung adenocarcinomas with mutations occurring at similar frequencies [57,58]. Moreover, cases of KRAS mutations were detected in both human lung cancers—NSCLC and SCLC [59,60].

PLGA-PEI-si-m/hVDAC1-B-induced metabolic reprogramming in the tumor resulted in inhibited tumor growth of 69–86%, as reflected from an MRI imaging and histology analysis of lung sections (Figure 2 and Figure 3). Interestingly, the remaining tumors in the lungs of the PLGA-PEI-si-m/hVDAC1-B-treated mice were localized at the surface borders (Figure 2C). A possible explanation for this is that less PLGA-PEI-si-m/hVDAC1-B reaches the borders of the lung. A longer treatment time or higher PLGA-PEI-si-m/hVDAC1 concentrations may overcome this.

Our results demonstrate that the two different concentrations (350 nM and 200 nM) of PLGA-PEI-si-m/hVDAC1-B that were used not only inhibited tumor growth (Figure 2 and Figure 3), but also decreased energy and metabolism in the tumor cancer cells (Figure 4). This is reflected not only in the decreased expression of VDAC1, but also of the glucose transporter Glut 1 and HK-I (Figure 4), shown to be overexpressed in many tumors [40,47].

In the lung sections the expression levels of SOX2 and ALDH1A1 were high in the non-tumor area, relative to the levels in the tumor in both the PLGA-PEI-si-NT and PLGA-PEI-si-m/hVDAC1-B-treated groups (Figure 5A,B).

ALDH1A1 is suggested to participate in the maintenance of CSCs and is known as a marker for lung CSCs, which have high tumor-initiating and self-renewing capabilities [61]. This enzyme could contribute to cancer, as well as regulation of metabolism and promotion of DNA repair. However, it was also found to act as a tumor suppressor in certain cancers [62].

ALDH1A1 expression is regulated by several epigenetic processes. It has been shown that key steps of epigenetics were altered including the expression of DNA methyltransferases (DNMTs), miR-29b, and HDAC1 in mouse lungs upon urethane treatment both before and after the development of tumors [33,63]. Thus, the decreased expression of ALDH1A1 in urethane-induced lung cancer may result from the urethane decreasing its expression via epigenetic alerions.

In support of these results, are the findings that in mice exposed to urethane, ALDH1 was abundantly expressed in normal lung tissue and was almost absent in lung tumors, demonstrated at the mRNA level with IHC staining [64].

SOX2 is a transcription factor that is essential for early mammalian development of many tissues and organs, and for the maintenance of both pluripotential embryonic stem cells and stem cells in multiple adult tissues [65]. Its high levels in the tumor-free area may be related to SOX2 playing an important role in lung epithelium. It is expressed in lung epithelial cells, both in the embryonic and adult trachea and in the airway/bronchiolar epithelium [66,67]. SOX2 is also important for the maintenance of Clara cells, ciliated cells, and goblet cells in the distal airway as Clara cell-specific deletion late in development leads to a cuboidal epithelium that lacks these columnar cell types [68].

However, the overexpression of SOX2 has been demonstrated in many different types of human cancers, and its expression promotes neoplastic progression by accelerating cancer cell proliferation, migration, invasion, and metastasis [69]. *SOX2* is overexpressed in human squamous cell lung tumors and in some adenocarcinomas, where it is proposed to play different roles [70]. In NSCLC, SOX2 suppresses radioimmune responses via activating the cGAS/STING signaling pathway [71].

The decrease in SOX2 expression levels in the urethane-induced tumors is not clear. Conceding that it has oncogenic and onco-suppressive activities [72], it is possible that in the urethane-derived tumors, SOX2 levels were decreased due to its onco-suppressive activities. In addition, it has been demonstrated that SOX2 is not universally required for the regulation of CSC-like properties, nor is it essential in lung adenocarcinoma cancer cells [73].

Finally, as discussed above for ALDH1A1, epigenetic modifications induced by urethane may downregulate SOX2 expression. Indeed, epigenetic silencing of SOX1 was reported in hepatocellular carcinomas [74].

### 4.2. Tumor VDAC1 Depletion as a Possible Treatment of Small and Non-Small Lung Cancer

In the majority of clinical cases, an SCLC diagnosis can be established based only on examining H&E-stained sections [46]. However, the immunostaining can provide a more accurate diagnosis. The biomarkers for SCLC include expression of NET markers [12] such as chromogranin A, synaptophysin, and CD56/NCAM, in addition to high expression levels of KI-67 and TTF-1, and negatively expressed p63 [10,13].

Because of their differences in behavior pattern and some genomic abnormalities, NSCLC and SCLC are treated in distinct ways [75,76,77]. In the early stages of NSCLC, radical surgery is the standard of care, with subsequent chemotherapy for complete resected NSCLC tumors [78]. However, in advanced stages, NSCLC is non-operable, and the aim is to slow down the progression of the malignancy. In this case, the first line treatment is chemotherapy and/or radiotherapy [79]. In addition, there are frequent cases of NSCLC transforming into SCLC during or after treatment [80,81].

SCLC remains one of the most aggressive types of lung cancer, with limited therapeutic options [82,83]. Although it is sensitive to chemotherapy and radiotherapy, it has a propensity for drug resistance. In this case, therapeutic options are limited and depend on the stage of SCLC, still with poor prognosis [84]. Thus, there is a strong need for a better treatment strategy for SCLC.

According to the reported studies [33,37], urethane induces a subtype of NSCLC, adenocarcinoma. It is possible that it can also induce a SCLC subtype, but this is difficult to distinguish. In order to verify the presence of SCLC, it is necessary to verify the expression of several specific markers.

According to our results (Figure 6), since urethane is known to induce mutations, it may contribute to genetic changes that lead to the development of both NSCLC and SCLC in A/J mice. This was demonstrated using several markers, such as KI-67 and synaptophysin staining. In the PLGA-PEI-si-NT tumor sections, two tumors that differed in their morphology and cell size were seen. One of them highly expressed KI-67 and synaptophysin (Figure 6), suggesting that this tumor is of SCLC type.

As PLGA-PEI-si-m/hVDAC1-B highly reduced the tumors induced in the lung cancer by urethane, we suggest that this treatment affects both NSCLC and SCLC. This was further demonstrated in a xenograft mouse model using an SCLC cell line (Figure 7 and Figure 8).

To support our conclusion from the urethane-induced lung cancer study showing that PLGA-PEI-si-m/hVDAC1-B inhibited small cell lung carcinomas, we established a xenograft SCLC mouse model using the human H69 cell line. Tumors were treated with NT-si or si-m/hVDAC1-B, with the latter inhibiting tumor growth (Figure 7C,D) via reprogramming the metabolism and inhibiting proliferation (Figure 7E–H and Figure 8A,B) and stemness (Figure 8C,D).

As the transformation of NSCLC into SCLC has been reported [80,81,85,86], our results present a treatment involving metabolic reprogramming via VDAC1 depletion, which inhibits the growth of tumors caused by NSCLC, SCLC, or their combination. These findings are very important, as SCLC is a very aggressive type of lung cancer with a poor prognosis.

Importantly, here, for the first time, it is demonstrated that si-m/hVDAC1-B reached the lung tumor, inhibited tumor growth of both small and non-small lung cancer and led to metabolism reprogramming and a decrease in the presence of CSCs, as reflected in the decreased expression of the CD44 and CD133 markers.

Immunotherapy has focused on augmenting cell-mediated immunity with a biological anti-tumor cell response, and involves the use of antibodies against cytotoxic T lymphocyte-associated antigen 4 (CTLA-4) and against the programmed death-1 (PD-1) and its ligands such as PD-L1 and PD-L2 [87]. These antibodies have been used to treat patients with advanced cancers such as melanoma; B cell chronic lymphocytic leukemia (CLL); colorectal, renal, and bladder cancers; and NSCLC [88]. Anti-PD-L1 has been used to treat NSCLC patients having no mutations in epidermal growth factor receptor (EGFR) or in anaplastic lymphoma kinase (ALK) and has a significant upregulated expression of PD-L1 and PD-L2. However, the response to immunotherapy in NSCLC patients is only about 30% [89]. Although this mode of immunotherapy looks promising, there are several obstacles to this treatment as cancer cells develop various mechanisms to escape an anticancer immune response [90]. Thus, new strategies for lung cancer treatment are required.

### 4.3. VDAC-1-Based Peptide, Retro-Tf-D-LP4 as a Potential Treatment for Lung Cancer

The VDAC1-based peptide, Retro-Tf-D-LP4, acts to interfere with the action of anti-apoptotic proteins overexpressed in cancer and to confer chemoresistance to apoptosis-inducing drugs [25]. In previous studies, we showed that in a xenograft mouse model derived from a A549 cell line of NSCLC, Retro-Tf-D-LP4 peptide treatment led to attenuation of tumor growth through the impairment of energy and metabolic processes, and induction of massive apoptotic cell death [25].

Here, we demonstrated that, in a urethane-induced lung cancer mouse model, the Retro-Tf-D-LP4 peptide reached the lung and attenuated tumor growth (Figure 9B), and also decreased the expression of metabolic-related enzymes such as VDAC1 and the TCA cycle enzyme citrate synthase (Figure 9C–E). In addition, the expression of the CSC marker, CD44, was also decreased (Figure 9E,F), suggesting, as shown previously in xenograft models and cell lines [26], that the peptide eliminates CSCs in urethane-induced tumors. These stem cells are characterized by pluripotentic and multipotentic stem cell markers [91]. This is important as CSCs have extraordinary resistance capabilities to conventional treatments.

## 5. Conclusions

In this study, using a carcinogen-induced lung cancer that is very close to human developed lung cancer, we demonstrated that siRNA against VDAC1 encapsulated in PLGA-PEI nanoparticles and VDAC1-based peptides developed in our lab [25,26] when given intravenously reached the lung tumors, effectively attenuating tumor growth and reversing their oncogenic properties. Moreover, our treatments eliminated tumors that represent either characterization of non-small lung or small lung cancer that require different treatments. This is most important as a transformation from non-small to small lung cancer has been reported, and here, we provide a treatment that affects both types of lung cancer. The results point to si-VDAC1 as a potential treatment for lung cancer.

## Figures and Tables

**Figure 1 cancers-16-02970-f001:**
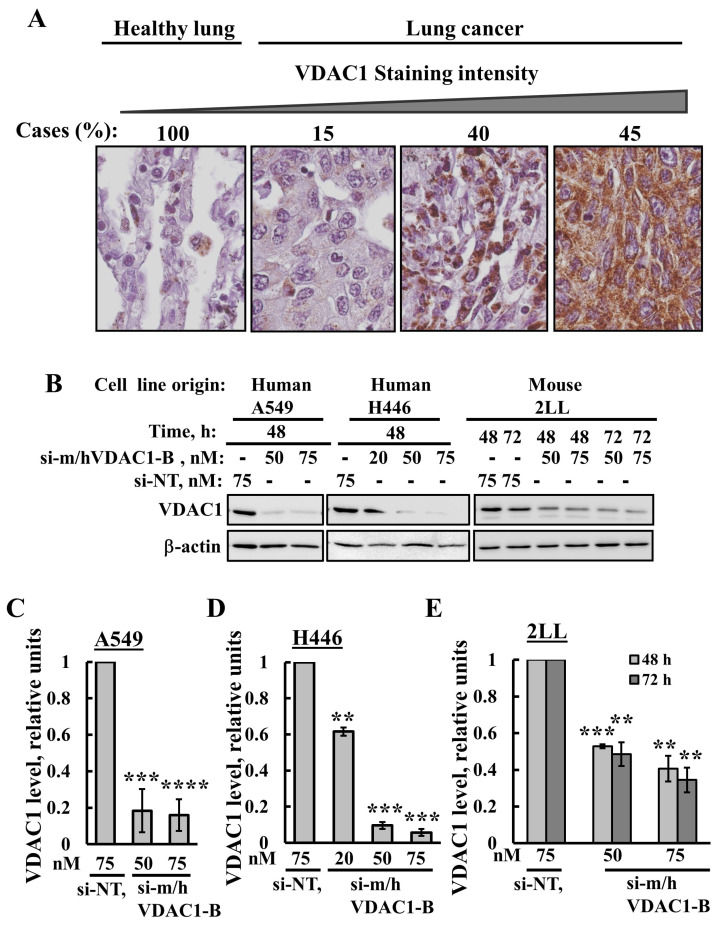
VDAC1 is overexpressed in lung cancer and VDAC1 expression is silenced in lung cancer cell lines using si-m/hVDAC1-B. (**A**) IHC staining with anti-VDAC1 antibodies was performed on lung tissue microarray slides (US Biomax). Representative images from sections of the NSCLC tissues (n = 31) and healthy pulmonary tissues (n = 10) with the percentage of sections stained at the intensity indicated by the scale above. Sections of the tissues were observed under an Olympus microscope, and images were taken at 600× magnification. (**B**) Lung cancer cell lines A549, H446 and 2LL were transfected with the indicated concentration of si-m/hVDAC1-B using silenFect transfection agent, as described in the Methods sections. Cells were harvested 48 or 72 h post transfection and subjected to immunoblotting using anti-VDAC1 antibodies. (**C**–**E**) Quantification of the VDAC1 expression levels in the three cell lines. Results present the mean ± SD (n = 3), ** *p* ≤ 0.01, *** *p* ≤ 0.001, **** *p* ≤ 0.0001. Original western blots are presented in Appendix A.

**Figure 2 cancers-16-02970-f002:**
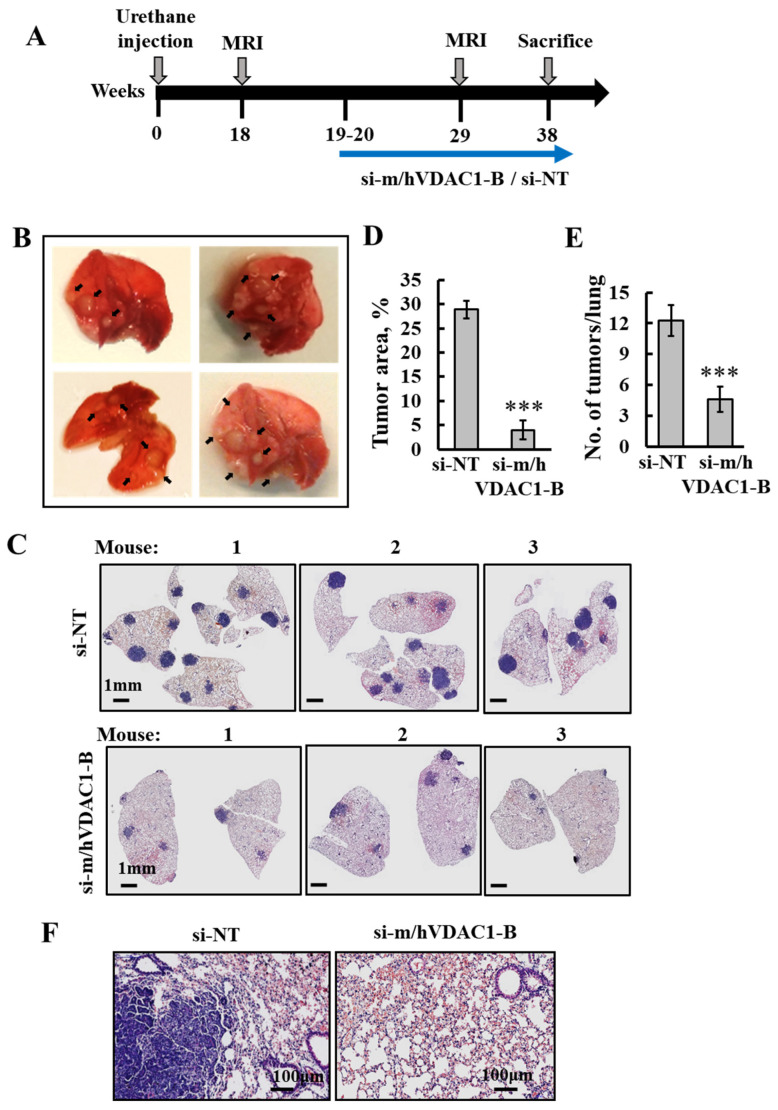
PLGA-PEI-si-m/hVDAC1-B treatment of urethane-induced lung cancer reduced tumor size and number. (**A**) Schematic illustration of the study design: A/J mice, 7–8 weeks old, were injected i.p. with urethane (1 mg/gr body weight), and tumor development and treatment are indicated. (**B**) Photo of lungs from mice sacrificed at week 19, after exposure to urethane, with the arrows pointing to tumors. Following tracking of tumor development with MRI at week 18, treatment with PLGA-PEI-si-m/hVDAC1 or PLGA-PEI-si-NT (200 nM blood concentration) was started at week 19, and performed three times a week until the mice were sacrificed at week 38. (**C**,**F**) Representative H&E staining of paraffin-embedded, formaldehyde-fixed sections of the lungs after treatment with PLGA-PEI-si-m/hVDAC1-B or PLGA-PEI-si-NT. (**D**,**E**) Quantitative analysis of tumor area presented as a % of the total section area (PLGA-PEI-si-NT, n = 7, PLGA-PEI-si-m/hVDAC1, n = 8) (**D**), and the number of tumors (PLGA-PEI-si-NT n = 4 and PLGA-PEI-si-m/hVDAC1-B, n = 4) (**E**). Results reflect the mean ± SEM, *** *p* ≤ 0.001. (**F**) Enlargement of a selected area from the lung shown in C.

**Figure 3 cancers-16-02970-f003:**
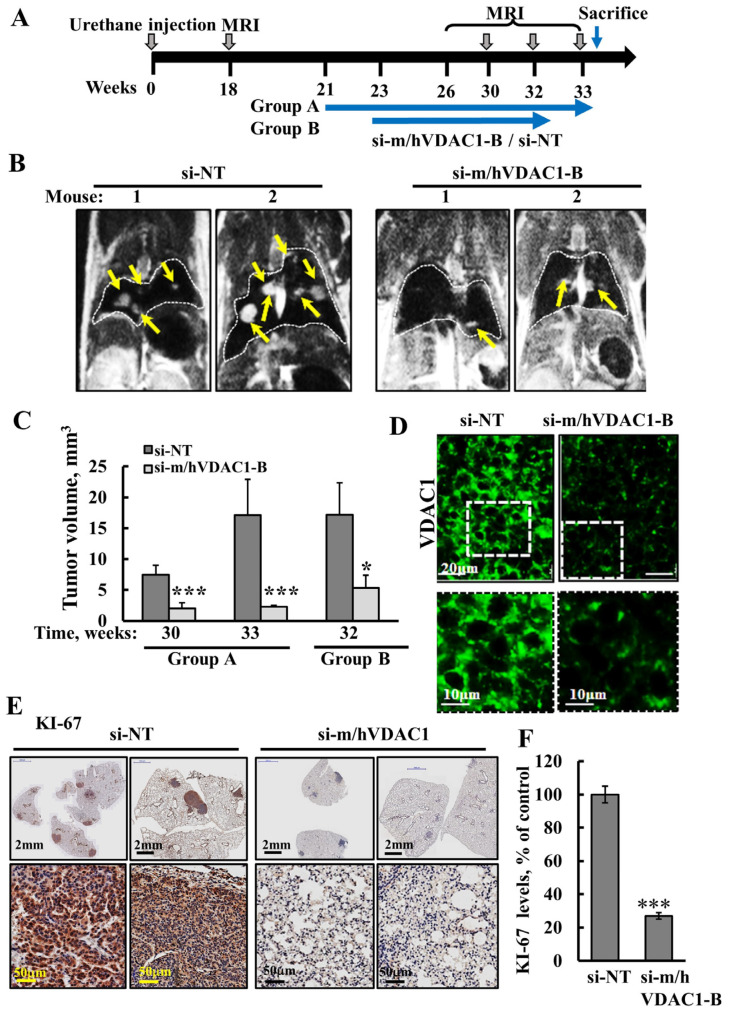
PLGA-PEI-si-m/hVDAC1-B treatment of urethane-induced lung cancer decreased VDAC1 expression level and inhibited tumor growth. (**A**) Schematic illustration of the study design: A/J mice, 7–8 weeks old, were injected i.p. with urethane (1 mg/gr body weight). Treatment with PLGA-PEI-si-m/hVDAC1 or PLGA-PEI-si-NT (350 nM) was started at week 21 (group A) or at week 23 (group B) and continued three times a week until the mice were sacrificed. MRI imaging was performed at weeks 18, 30, 32, and 33. Mice were sacrificed at week 33 (group A) and week 32 (group B). (**B**) Representative MRI images at week 33 (group A) of PLGA-PEI-si-NT- and PLGA-PEI-si-m/hVDAC1-B-treated mice. The arrows point to tumors. (**C**) Quantification of tumor volume from MRI images of group A at week 30 (PLGA-PEI-si-NT, n = 6, PLGA-PEI-si-m/hVDAC1-B, n = 7) and week 33 (PLGA-PEI-si-NT, n = 5, PLGA-PEI-si-m/hVDAC1-B, n = 5) and for group B at week 32 (PLGA-PEI-si-NT, n = 3, PLGA-PEI-si-m/hVDAC1-B, n = 4). (**D**) IF staining of VDAC1 in lung sections from PLGA-PEI-si-m/hVDAC1-B and PLGA-PEI-si-NT-treated mice. The squares point to the area enlarged below. (**E**,**F**) IHC staining for KI-67 using specific antibodies of PLGA-PEI-si-NT- and PLGA-PEI-si-m/hVDAC1-B-treated mice (**E**). For the sections, the lungs with tumors are shown with enlarged area below. KI-67 staining intensity was analyzed quantitatively (n = 7) (**F**). Results present the mean ± SEM, * *p* ≤ 0.05, *** *p* ≤ 0.001.

**Figure 4 cancers-16-02970-f004:**
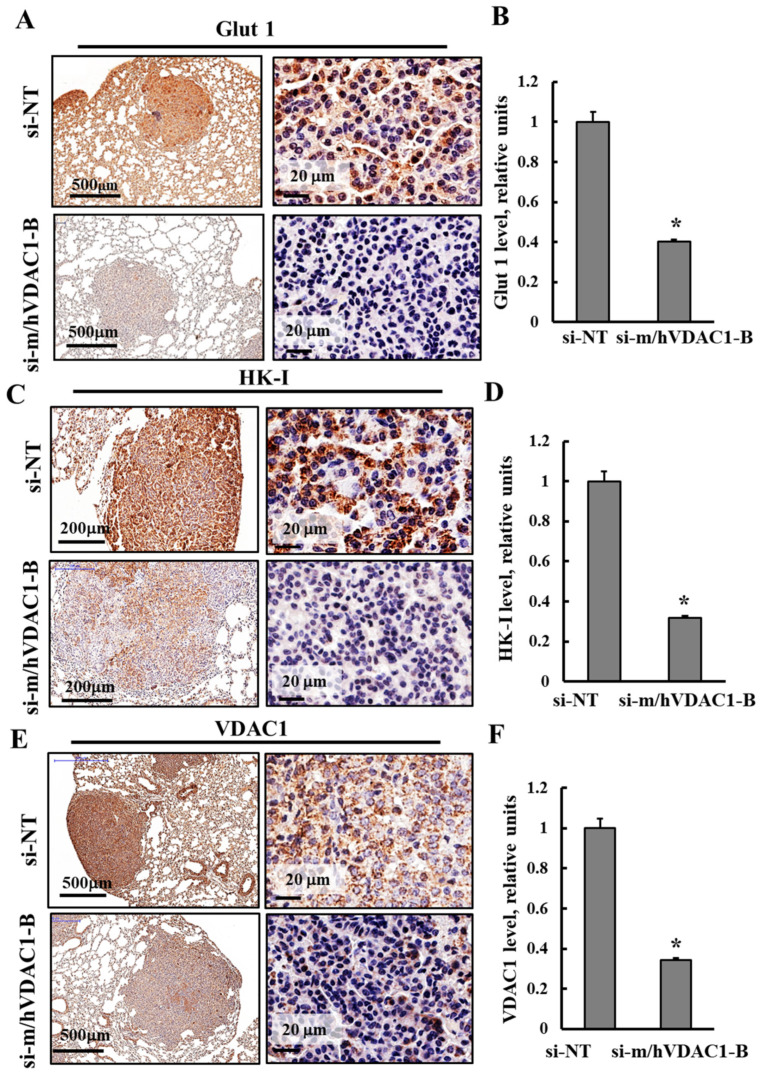
PLGA-PEI-si-m/hVDAC1-B treatment reduced expression of energy- and metabolism-related proteins. Representative IHC staining of lung sections of PLGA-PEI-si-m/hVDAC1-B- and PLGA-PEI-si-NT-treated mice using specific antibodies against Glut 1 (**A**,**B**), HK-I (**C**,**D**) and VDAC1 (**E**,**F**), shown in two different magnifications. Staining intensity was quantified as described in the Methods section. Results present the mean ± SEM (n = 5), * *p* ≤ 0.05.

**Figure 5 cancers-16-02970-f005:**
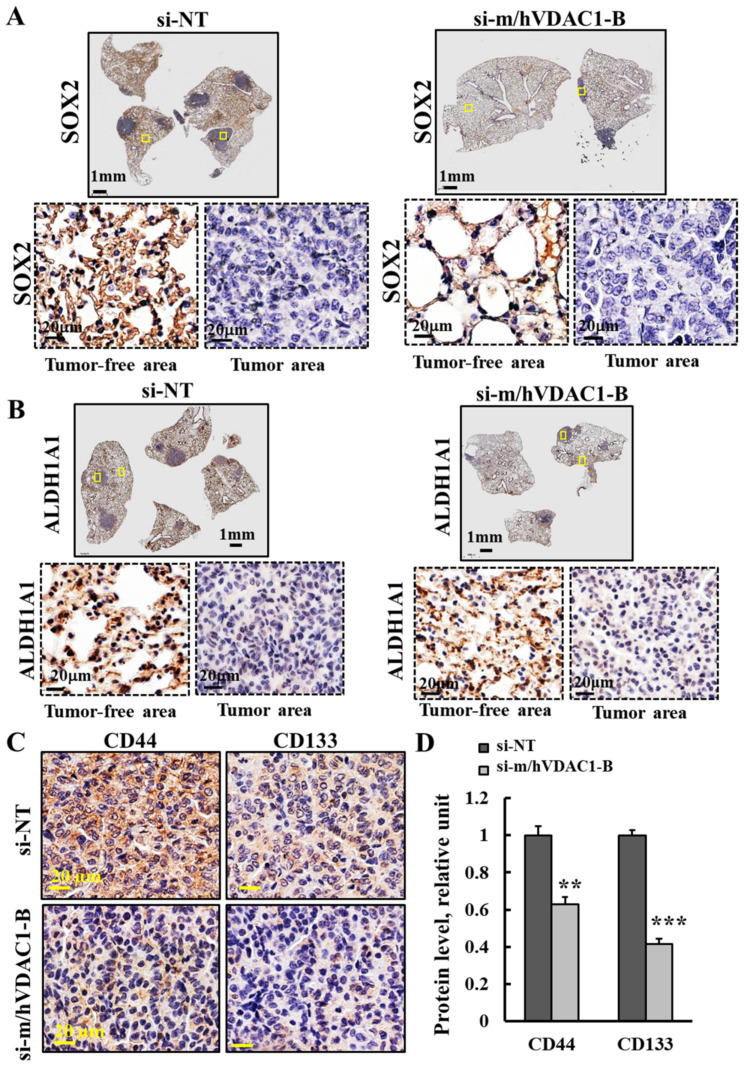
Effect of PLGA-PEI-si-m/hVDAC1-B treatment on the expression of CSC-related markers. (**A**,**B**) Representative lung sections IHC stained with specific antibodies against SOX2 (**A**) and ALDHA1 (**B**), showing enlarged areas from tumor-free and tumor tissues. (**C**,**D**) Representative IHC staining of stem cell markers CD44 and CD1133 in PLGA-PEI-si-NT- and PLGA-PEI-si-m/hVDAC1-B-treated mice (**C**) and their quantified staining intensity (**D**). Results reflect the mean ± SEM (n = 3), ** *p* ≤ 0.01, *** *p* ≤ 0.001.

**Figure 6 cancers-16-02970-f006:**
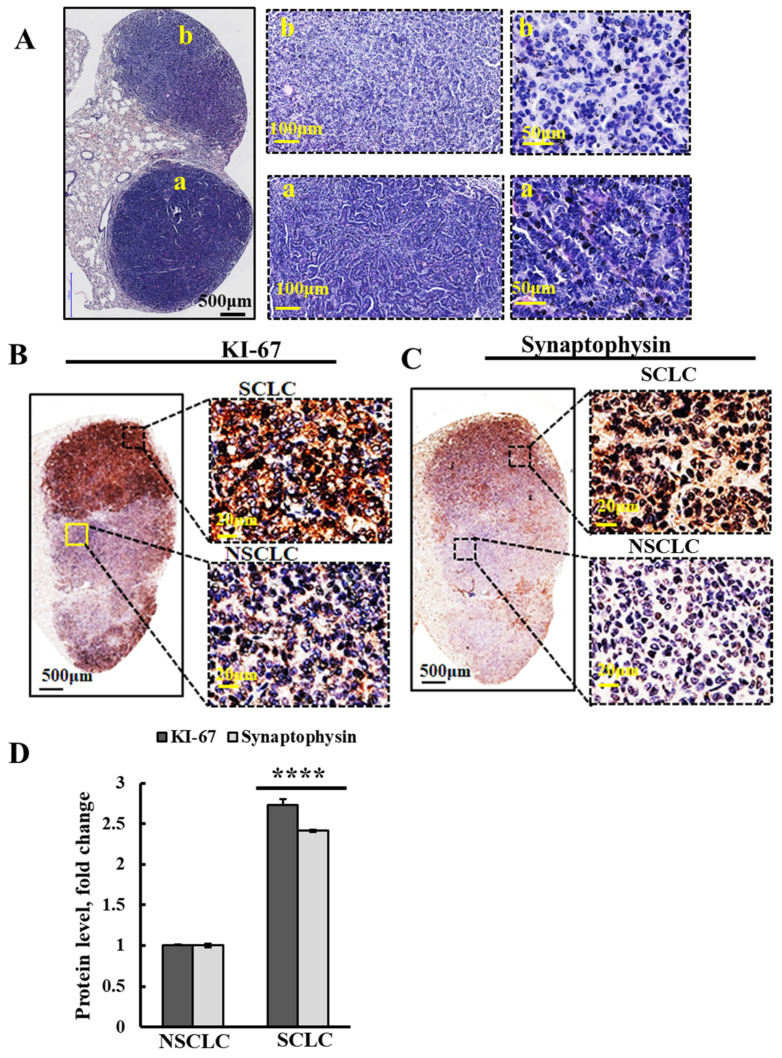
Urethane treatment of A/J mice may induce both small and non-small lung cancer. (**A**) Representative H&E staining of formaldehyde-fixed paraffin-embedded sections from A/J mice lungs treated with PLGA-PEI-si-NT, showing two different morphologies as visualized in images at three different magnifications. (**B**,**C**) Representative lung sections of PLGA-PEI-si-NT-treated mice, IHC stained using specific antibodies against KI-67 (**B**) and synaptophysin (**C**), known to be highly expressed in SCLC. The squares show the areas from the high and low IHC-stained tumors that were magnified 25-fold in the images shown on the right. (**D**) Quantitative analysis of the indicated proteins’ staining intensities. Results reflect the mean ± SEM (n = 3), **** *p* ≤ 0.0001.

**Figure 7 cancers-16-02970-f007:**
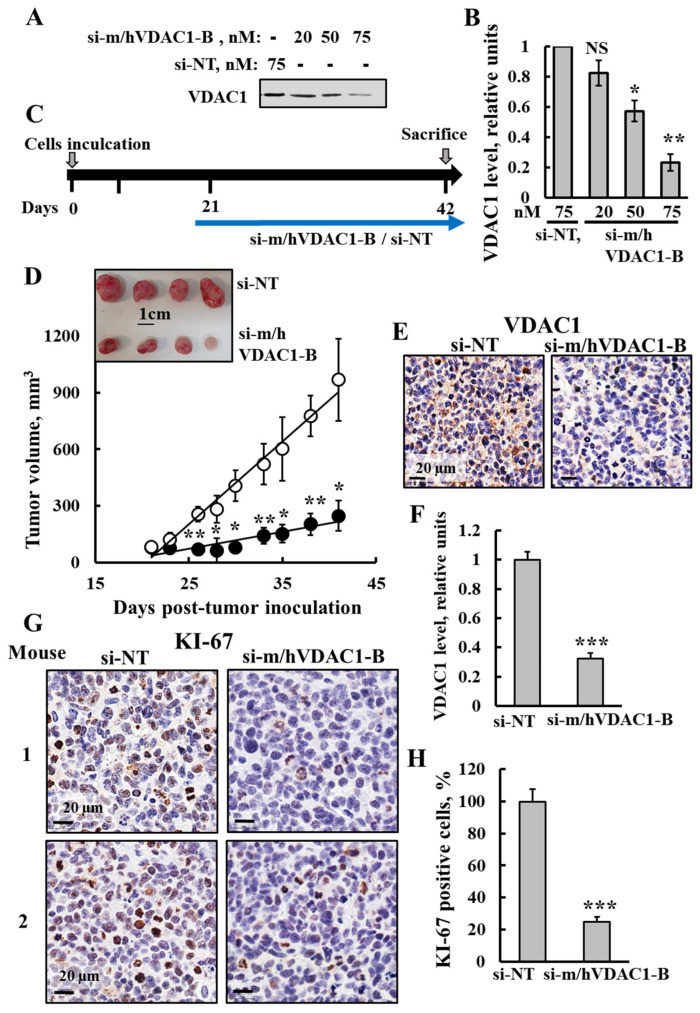
si-m/hVDAC1-B tumor treatment inhibited tumor growth, reduced VDAC1 expression and cell proliferation in an SCLC xenograft mice model. (**A**,**B**) SCLC H69 cells were transfected with the indicated concentration of si-NT or si-m/hVDAC1-B using silenFect transfection agent, as described in the Methods sections. Cells were harvested 48 h post transfection and subjected to immunoblotting using anti-VDAC1 antibodies (**A**) and VDAC1 expression level quantification (**B**). (**C**) Schematic illustration of the study design: H69 cells (2 × 10^6^ cells/mouse) were inoculated into Athymic (C57BL/6JOlaHsd) 6-week-old male mice. When the tumor volume was 50–80 mm^3^ (at day 21), the mice were sub-divided into two groups: si-m/hVDAC1-B (200 nM) and si-NT (200 nM) -treated xenografts, injected three times a week and the mice were scarified at day 42. (**D**) Tumor size was measured (using a digital caliper) and tumor volume was calculated with the average presented as the mean ± SEM.; si-m/hVDAC1-B (●, n = 5) and si-NT (o, n = 4). A photograph of the tumors excised from sacrificed mice is also shown (inset). (**E**,**F**) Representative IHC-stained tumor sections of si-m/hVDAC1-B- and si-NT-treated mice using specific antibodies against VDAC1 (**E**) and their quantified staining intensity (**F**). (**G**,**H**) Representative IHC-stained tumor sections of si-m/hVDAC1-B- or si-NT-treated tumors using specific antibodies against KI-67 (**G**). Quantitative analysis of KI-67 positive cells (**H**). Results reflect the mean ± SEM (n = 4–5), * *p* ≤ 0.05, ** *p* ≤ 0.01, *** *p* ≤ 0.001. Original western blots are presented in Appendix A.

**Figure 8 cancers-16-02970-f008:**
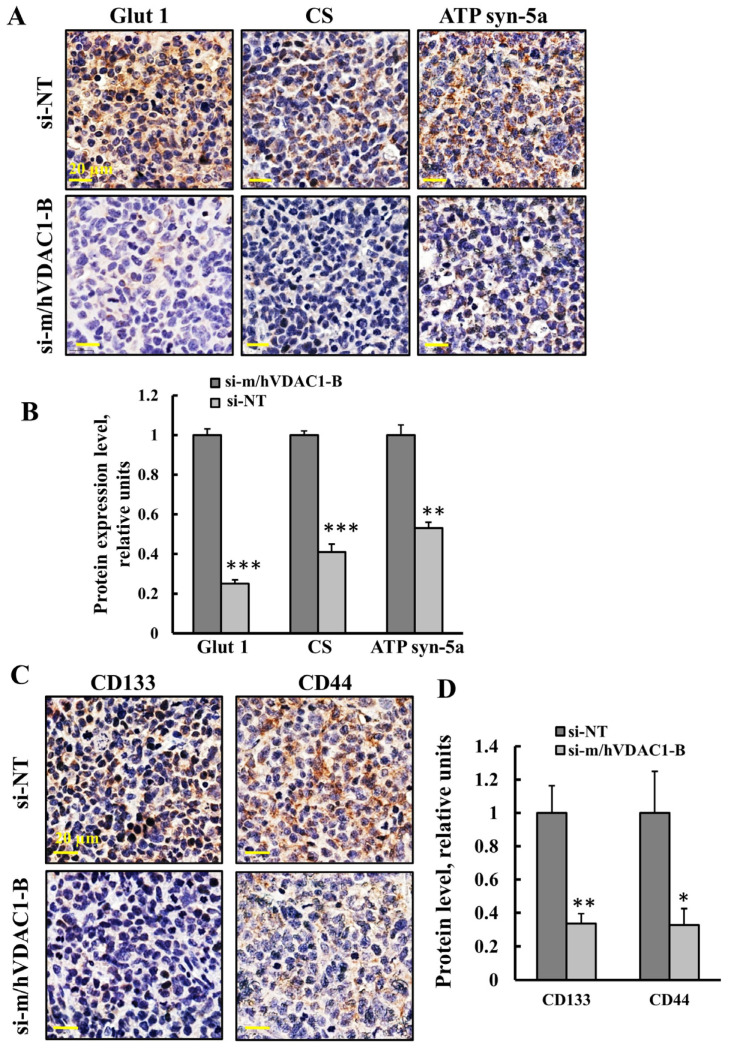
Tumor treatment with si-m/hVDAC1-B reduced the expression of metabolism- and CSC-related proteins in an SCLC xenograft mouse model. (**A**,**B**) Representative IHC stained tumor sections of si-m/hVDAC1-B- and si-NT-treated mice with specific antibodies for Glut 1, CS, and ATP synthase 5a (**A**) and their quantified staining intensity (**B**). (**C**,**D**) Representative IHC stained tumor sections from si-m/hVDAC1-B- and si-NT-treated tumors, using specific antibodies against CD44 and CD133 (**C**) and their quantified staining intensity (**D**). Results represent the mean ± SEM (n = 3), * *p* ≤ 0.05, ** *p* ≤ 0.01, *** *p* ≤ 0.001.

**Figure 9 cancers-16-02970-f009:**
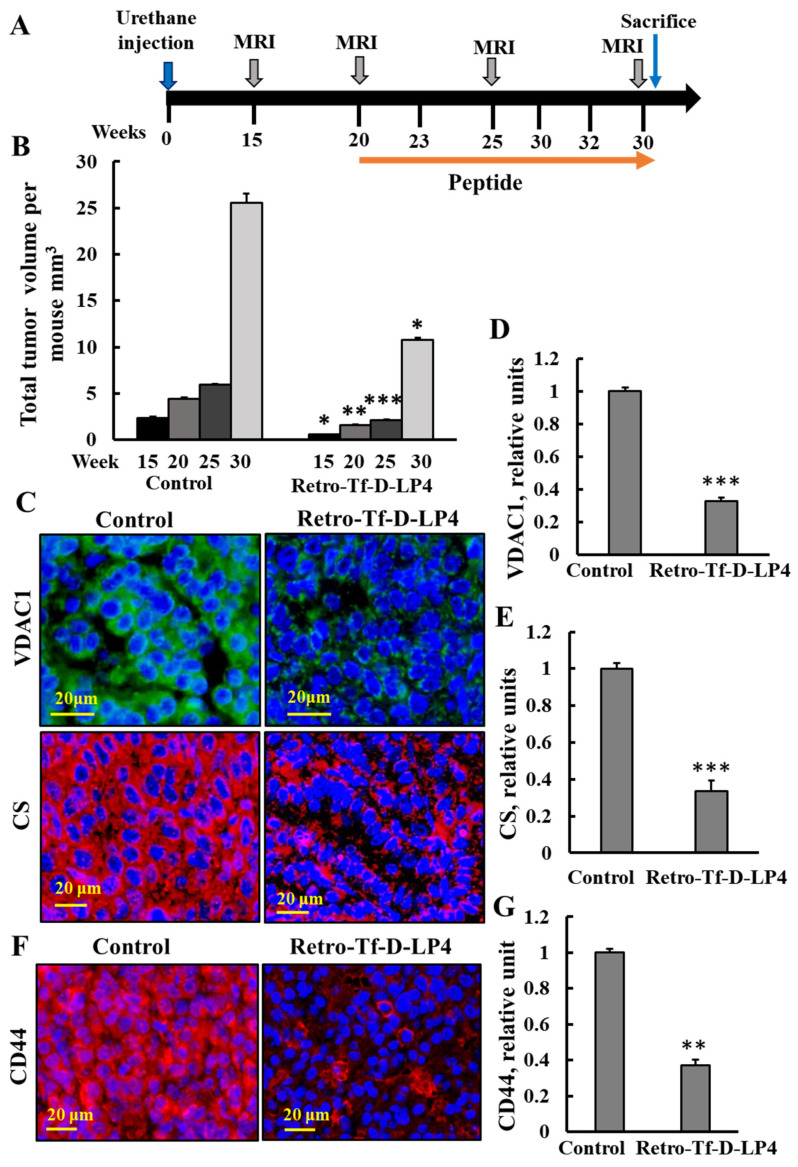
Urethane-induced lung cancer mice, treated with Retro-Tf-D-LP4 peptide, showed inhibited tumor growth and decreased expression levels of metabolism-related enzymes and CSC markers. (**A**) Schematic illustration of the study design: A/J mice, 7–8 weeks old, were injected i.p. with urethane (1 mg/gr body weight). At week 15, tumors were evident by MRI imaging, and i.v. treatment with the Retro-Tf-D-LP4 peptide (18 mg/Kg) or with DMSO (control) was started at week 20, and continued three times a week until the mice were sacrificed at week 30. (**B**) MRI tracking of tumors was carried out at weeks 15, 20, 25, and 30. (**C**–**E**) Representative lung tumor sections of Retro-Tf-D-LP4 and DMSO-treated mice, IF-stained using specific antibodies against VDAC1 and CS (**C**), and their quantified staining intensity is shown (**D**,**E**). (**F**) Representative tumor sections from lungs of mice treated with urethane and with Retro-Tf-D-LP4 or DMSO (control) IF stained using specific antibodies against CD44 and quantified staining intensity (**G**). Results reflect the mean ± SEM (R-Tf-D-LP4 peptide, n = 5; control n = 4), * *p* ≤ 0.05, ** *p* ≤ 0.01, *** *p* ≤ 0.001.

**Table 1 cancers-16-02970-t001:** Antibodies used in this study.

Antibody	Source and Cat. No.	Dilution		
		IHC	WB	IF
Rabbit polyclonal anti-VDAC1	ab15895, Abcam, Cambridge, UK	1:400	1:15,000	1:500
Rabbit polyclonal anti-KI-67	ab15580, Abcam, Cambridge, UK	1:500		
Rabbit polyclonal anti-Glut 1	ab652, Abcam, Cambridge, UK,	1:500		
Rabbit polyclonal anti-ATP5A	ab151229, Abcam, Cambridge, UK	1:500		
Mouse monoclonal anti-SOX2	ab171380, Abcam, Cambridge, UK	1:200		
Rabbit polyclonal anti-CD44	ab157107, Abcam, Cambridge, UK	1:800		1:1000
Rabbit monoclonal anti-HK-I	ab150423, Abcam, Cambridge, UK	1:100		
Rabbit polyclonal anti-CD133	ab19898, Abcam, Cambridge, UK	1:200		
Rabbit monoclonal anti-synaptophysin	ab32127, Abcam, Cambridge UK	1:500		
Rabbit polyclonal anti-citrate synthetase	ab96600, Abcam, Cambridge, UK	1:500		1:500
Rabbit monoclonal anti-ALDH1	ab52492, Abcam, Cambridge, UK	1:50		
Mouse monoclonal anti-β-actin	MAB1501, Millipore, Billerica, MA		1:40,000	
Donkey anti-Mouse (Alexa Fluor 488)	ab150109, Abcam, Cambridge, UK			1:750
Goat anti-Rabbit (Alexa Fluor 555)	ab150086, Abcam, Cambridge, UK			1:850
Goat anti-Rabbit HRP	W4018, Promega, Wisconsin	1:1000	1:15,000	
Donkey anti-Mouse HRP	ab98799, Abcam, Cambridge, UK	1:1000		

## Data Availability

No new data were created or analyzed in this study. Data sharing is not applicable to this article.

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
