# Peer review of "Mitochondrial VDAC1 Silencing in Urethane-Induced Lung Cancer Inhibits Tumor Growth and Alters Cancer Oncogenic Properties"

_cancers, 2024, doi:10.3390/cancers16172970_

Round 1

Reviewer 1 Report

Comments and Suggestions for Authors

The manuscript by Melnikov et al. presents compelling evidence that siRNA against VDAC1, when encapsulated in PLGA-PEI nanoparticles along with VDAC1-based peptides, can be administered intravenously and successfully target lung tumors. This targeted approach not only inhibits tumor growth but also reverses their oncogenic properties. This study demonstrated the effectiveness of the VDAC1-based peptide, Retro-Tf-D-LP4, in inhibiting tumor growth and reducing the expression of proteins associated with metabolism and cancer stem cells.

The manuscript is well-written, adding a vital layer of information about lung cancer therapy.

I have a few comments for polishing the manuscript.

 Major Comments

  1. The authors should discuss the molecular mechanisms behind how PLGA-PEI-si-m/h VDAC1-B works in a urethane-induced lung cancer model. They did discuss some metabolism-related proteins (GLUT1/HK-I), but that does not tell the whole story.  To shed light on this, the authors should perform differential gene expression analysis or comparative proteomics.
  2. The authors showed that urethane induced both SCLC and NSCLC. This is very interesting. There are many papers that show the transformation of NSCLC to SCLC. It would be intriguing to investigate the role of urethane in this process.
  3. SCLC is dependent on OXPHOS, the authors should test some OXPHOS markers in the presence of siRNA and peptide. The authors should further test whether siRNA or peptide used the same mechanisms to inhibit both SCLC and NSCLC.
  4. Did the authors investigate the combined effect of siRNA-VDAC1/peptide and immune check inhibitors on xenograft models?

Minor Comments

  1. The statistical data should be presented as mean ± SD.
  2. The western blot images should be represented graphically with three independent replicates.

Author Response

Reviewer 1

Comments and Suggestions for Authors

We thank this reviewer for pointing out that: "The manuscript is well-written, adding a vital layer of information about lung cancer therapy". The valuable feedback has been used to revise our manuscript in accordance with the referee's specific comments.

 Major Comments

  1. The authors should discuss the molecular mechanisms behind how PLGA-PEI-si-m/h VDAC1-B works in a urethane-induced lung cancer model. They did discuss some metabolism-related proteins (GLUT1/HK-I), but that does not tell the whole story.  To shed light on this, the authors should perform differential gene expression analysis or comparative proteomics.

Ans. 1: In the introduction, we have  detailed the function of VDAC1 in cancers (lines 67-75)  and discussed the effects of its silencing  on the tumor development and reprograming (lines 76-90):  "Previously, we demonstrated that silencing VDAC1 expression using si-RNA reduced cellular ATP levels, cell proliferation of a panel of cell lines regardless of cell origin, and mutation status [18-21, 23, 24, 29, 30]. In established mouse models, silencing VDAC1 expression  inhibited solid tumor development and growth in cervical, lung, bladder, and breast cancers, mesothelioma and glioblastoma tumors resulted in metabolic rewiring, leading to a reversal of their oncogenic properties, which  inhibited tumor growth, invasivity, stemness,  epithelial-to-mesenchymal transition  (EMT), and angiogenesis [18-21, 23, 24, 29, 30].  The treated metabolism reprogrammed residual “tumor” showed decreased proliferation, an altered tumor-microenvironment [21], and epigenetic-related enzymes and factors [31]. These epigenetic modifications point to an interplay between metabolism and epigenetics and can explain the changes in the expression of thousands of genes, transcription factors, and proteins, and lead to cell differentiation toward less malignant lineages [17-21]. Thus, VDAC1 is a key regulator of metabolic and energy reprogramming that disrupts cancer energy and metabolism homeostasis by downregulating VDAC1, making it a potential anti-cancer therapy strategy [17-24, 32]".

   Similarly, in the first paragraph of the Results section we have indicated:" Previously, we showed that silencing VDAC1 expression using human-specific siRNA (si-hVDAC1) resulted in the inhibition of cancer cell growth, both in vitro and in vivo in mouse intracranial and xenograft models of human glioblastoma, lung cancer, triple negative breast cancer, mesothelioma, and bladder cancers [17-21, 23, 24, 29, 30]. The results of these studies showed that tumor metabolic rewiring results in a reversal of their oncogenic properties, leading to inhibited tumor growth, invasivity, stemness, EMT, and angiogenesis [17-21, 23, 24, 29, 30].

  As suggested, we have added a brief paragraph to the Discussion section on metabolic reprogramming and its modulation of tumor oncogenic properties demonstrated in several mouse cancer models (lines 602-614).

We performed both NGS and proteomics analyses on control and si-VDAC1-treated tumors derived from GBM and lung cancer. The results revealed significant changes in the expression of thousands of genes, transcription factors, and proteins associated with epigenetic alterations, leading to cell differentiation towards less malignant lineages. These findings are now included in the revised text (lines 85-87).

We would like to emphasize that the focus of this study is on a chemically-induced lung cancer model that closely mimics the clinical progression of lung cancer, as it develops over a long period of time through mutations such as KRAS (add lines 94-97). We have demonstrated that, also in this model, inhibition of VDAC1 expression leads to reprogramming of cancer cell metabolism and elimination of cancer stem cells (CSCs), consistent with findings previously observed in SC mouse models. [17-21, 23, 24, 29, 30].

  1. The authors showed that urethane induced both SCLC and NSCLC. This is very interesting. There are many papers that show the transformation of NSCLC to SCLC. It would be intriguing to investigate the role of urethane in this process.

Ans. 2: We are excited to report that we successfully inhibited both SCLC and NSCLC by silencing VDAC1 expression, which is particularly significant given the findings related to the transformation of NSCLC into SCLC [86,87].

However, we are unclear about the reviewer's comment: "It would be intriguing to investigate the role of urethane in this process."  We added (lines 715-716) that since urethane is known to induce mutations, it may contribute to genetic changes that lead to the development of both NSCLC and SCLC.

  1. SCLC is dependent on OXPHOS, the authors should test some OXPHOS markers in the presence of siRNA and peptide. The authors should further test whether siRNA or peptide used the same mechanisms to inhibit both SCLC and NSCLC.

Ans. 3: Regarding SCLC metabolism, being OXPHOS.  It is indeed primarily dependent on OXPHOS, which requires the activity of the tricarboxylic acid (TCA) cycle. We have analyzed key components of this pathway, including citrate synthase (CS) and ATP synthase, as shown in Figure 8.

Regarding the mechanism of action for siRNA and peptide inhibition in both SCLC and NSCLC: The mechanisms of action for both siRNA and peptides are well-characterized and are not specific to any particular cancer cell line or cancer type, regardless of their origin or mutational status, as demonstrated in several of our publications [17-21,23,24,29,30] [25-27]. Therefore, it is most likely that their mode of action in SCLC and NSCLC is similar.

  1. Did the authors investigate the combined effect of siRNA-VDAC1/peptide and immune check inhibitors on xenograft models?

No, we did not perform that analysis, but it is a good idea, and we may consider it for future research.

Minor Comments

  1. The statistical data should be presented as mean ± SD.

As in all our studies over the past 20 years, we present standard error (SE) for comparisons and consistency. We prefer to use SE, as the p-value is independent of whether standard deviation (SD) or SE is used.

  1. The western blot images should be represented graphically with three independent replicates.

As suggested, we have now presented the band staining intensities graphically. In Figure 1, please refer to panels A, C, and D. In Figure 7, see panel B.

Reviewer 2 Report

Comments and Suggestions for Authors

Melnikov et al. have presented a delicate work demonstrating the role mitochondrial protein VDAC1 plays in NSCLC and SCLC metabolism both in vitro and in vivo, that its elevation could lead to the malignancy of the tumor whereas inhibition of such protein reduces tumor growth. This work explores the potential to develop a novel therapeutic pathway that could lead to the reduction of cellular metabolism and malignant growth. However, there are a few issues the authors could address before the acceptance of this work.

1)      Since VDAC1 is a mitochondrial protein, when performing a quantitative analysis of the expression of this protein, a mitochondria-specific housekeeping protein should be used as the control, such as TOMM20. Otherwise, it is hard to attribute to VDAC1-specific mechanisms within the mitochondria rather than some general changes on the mitochondria level.  

2)      The authors claimed that the dysregulation of VDAC1 in cancer cells caused metabolism reprogramming, hence contributing to the oncogenic activity. However, evidence is lacking to support such a claim. Please perform a live-cell mitochondria function assay or mitochondrial stress test to support this key point of your research with relevant data. For example, does the knockdown of VDAC reduce mitochondria respiration from the baseline level?

3)      Mitochondrial morphology is highly dynamic and subject to change to meet cellular energy demand. What do the authors think of the fusion-fission dynamics of mitochondria in NSCLC and SCLC, and how they are impacted by VDAC1 overexpression and knockdown? Are there experiments needed?

4)      Please show western blot results to quantify protein expression (GLUT1, HK-1, VDAC1) in Figure 4. IHC is not a quantitative way to show protein expression. It may be acceptable for surface markers, but not for key intracellular proteins.  

Author Response

Reviewer 2

 Comments and Suggestions for Authors

We thank this reviewer for pointing out that: "This work explores the potential to develop a novel therapeutic pathway that could lead to the reduction of cellular metabolism and malignant growth. The valuable feedback has been used to revise our manuscript in accordance with the referee's specific comments.

As we attached some figures, we suggest to this reviewer to use the attached PDF file

  1. Since VDAC1 is a mitochondrial protein, when performing a quantitative analysis of the expression of this protein, a mitochondria-specific housekeeping protein should be used as the control, such as TOMM20. Otherwise, it is hard to attribute to VDAC1-specific mechanisms within the mitochondria rather than some general changes on the mitochondria level.  

It is well-established that the expression of many proteins is elevated in cancer cells. In our previous study (Ref. 23) shown below (in the PDF), we showed the expression of VDAC1 but not of citrate synthase (CS), a mitochondrial protein is highly elevated in most cancer cell lines tested.

Conducting a similar experiment using patient tissue samples involves ordering slides, performing staining procedures, and analyzing the results, which is estimated to take approximately 2 months. However, we believe this experiment is not critical to our study’s core findings. The key point of our research is that cancer cells exhibit overexpression of VDAC1, and that inhibiting VDAC1 expression with siRNA effectively reduced tumor growth and eliminated its oncogenic properties.

  1. The authors claimed that the dysregulation of VDAC1 in cancer cells caused metabolism reprogramming, hence contributing to the oncogenic activity. However, evidence is lacking to support such a claim. Please perform a live-cell mitochondria function assay or mitochondrial stress test to support this key point of your research with relevant data.

 For example, does the knockdown of VDAC reduce mitochondria respiration from the baseline level?

 Ans- Our statement that down-regulation of VDAC1 expression levels in cancer cells leads to metabolism reprogramming, is based on multiple studies in cell-based assays and various cancer mouse as summarized in the Introduction (lines (lines 67-90), Results (lines 289-295) and Discussion (lines, 602-614).   

Studying isolated mitochondrial function alone does not fully capture metabolic reprogramming, as glycolysis cannot be assessed in such assays.

Using cells in culture, we demonstrated that silencing VDAC1 expression results in decreased mitochondrial membrane potential and ATP production, as shown in the presented figure from Ref. 17 (can be seen in the attached PDF).  

Moreover, studying metabolic reprogramming is more accurately conducted within the tumor context, including its microenvironment, which affects oxygen supply. As demonstrated, silencing VDAC1 significantly reduced angiogenesis, thereby impacting oxygen supply to the cancer cells.

  1. Mitochondrial morphology is highly dynamic and subject to change to meet cellular energy demand. What do the authors think of the fusion-fission dynamics of mitochondria in NSCLC and SCLC, and how they are impacted by VDAC1 overexpression and knockdown? Are there experiments needed?

Ans- Indeed, mitochondrial morphology and fusion-fission dynamics are both influenced by cellular energy and impact the cell's energy sources. Mitochondrial fusion facilitates effective oxidative phosphorylation (OXPHOS), ATP production, and the exchange of matrix content, while fission promotes glycolysis, reactive oxygen species (ROS) production, and mitophagy (Wu et al., Mitochondrial fusion–fission dynamics and its involvement in colorectal cancer, Mol Oncol. 2024 May; 18(5): 1058–1075).

Given that VDAC1 regulates both OXPHOS and glycolysis in cancer cells, its silencing is expected to affect fusion-fission dynamics. However, to date, there are no reported studies on the impact of VDAC1 depletion on these dynamics, except for Drosophila (Drosophila Porin/VDAC Affects Mitochondrial Morphology, PLoS One. 2010; 5(10): e13151).

 This is an interesting area for future research but will be addressed in a separate study.

  1. Please show western blot results to quantify protein expression (GLUT1, HK-1, VDAC1) in Figure 4. IHC is not a quantitative way to show protein expression. It may be acceptable for surface markers, but not for key intracellular proteins.  

Ans - In immunohistochemistry (IHC) using fixed and paraffin-embedded tissue, first the tissues were when subjected to hexane and ethanol extraction, thus all proteins are exposed to interaction with antibodies. The detection of primary antibodies bound to target proteins is obtained using secondary antibodies conjugated to horseradish peroxidase (HRP). When all samples are incubated with the secondary antibodies simultaneously and for the same duration, the staining intensity, which reflects HRP activity, is proportional to the amount of bound primary antibody.

As indicated, the slides were scanned using a Panoramic microscope, and the intensity was analyzed using the ImageJ program. Quantification of IHC for cellular proteins is a well-established method and is widely used in many studies (see: Alexandra R. Crowe and Wei Yue, "Semi-quantitative Determination of Protein Expression Using Immunohistochemistry Staining and Analysis," Bio Protoc. 2019 Dec 20; 9(24): e3465).

Reviewer 3 Report

Comments and Suggestions for Authors

The paper titled "Mitochondrial VDAC1 silencing in urethane-induced lung cancer inhibits tumor growth and alters cancer oncogenic properties" demonstrates the role of VDAC1 silencing in lung cancer cell lines. Followed by siRNA treatment, a lung cancer A/J mice model treated with urethane was subjected to PLGA-PEI-si-m/hVDAC1-B injections to perform immunohistochemistry or immunofluorescence assays. Xenograft mice (C57BL/6JOlaHsd) models were generated using H69 cells suspension injections subcutaneously and checked for tumor growth inhibition after si-m/hVDAC1-B treatment. The study is well-framed, and the results have been reflected clearly. However, there are a few corrections that need to be made:

1.      In the materials and methods section 2.6. (Urethane-induced lung tumor mice model), two groups of mice have been mentioned, but the number of mice used in each group has been reflected roughly as 4-7 mice/group. It would be better if there was a clear explanation of the study design, groups of mice, number of mice, their age, at which time point they were euthanized, etc.

2.      Similarly, in section 2.8 (Xenograft mouse model), each group reflects 7 mice/group, but did all of them survive after treatment, or did a few die? Such information should be clearly reflected.

3.      A chart or a tabular representation of the entire study design with a timeline for sections 2.6 and 2.8 should be given reflecting the no. of mice used, their initial and final weights, day of treatment, day of euthanization, etc., for clarity and better understanding.

4.      In Figure 1, part B, one of the blots of mouse cell line 2LL reflecting the VDAC1 expression level has some impurity as well because the blots reflect double lines. Check and, if necessary, change that blot image.

5.      The B and D parts of Figure 3 need to be updated with a better-resolution image. Also, the B and C parts of Figure 6 need to be updated with a better-resolution image.

6.      In Figure 7, part B, two of the shaded circles in black don't have any error bars. It needs to be rechecked and modified. In Figure 8, part B, the significance shown with stars has been shown vertically. Please maintain uniformity while using signs and symbols for significance, it should be horizontal everywhere. In Figure 9, part D, there is a misalignment in the significance shown, which needs to be corrected.

Author Response

Reviewer 3

 Comments and Suggestions for Authors

 We thank this reviewer for pointing out that: "The study is well-framed, and the results have been reflected clearly". The reviewer's valuable feedback has been used to revise our manuscript in accordance with the referee's specific comments.

  1. In the materials and methods section 2.6. (Urethane-induced lung tumor mice model), two groups of mice have been mentioned, but the number of mice used in each group has been reflected roughly as 4-7 mice/group. It would be better if there was a clear explanation of the study design, groups of mice, number of mice, their age, at which time point they were euthanized, etc.

Ans. As suggested, we have revised the information in this section to accurately present the protocols for the three experiments. MRI was performed three times during the experiment: before starting the treatment, several weeks after the treatment commenced, and just before termination (mice euthanasia).

  • Experiment I: Comparison of the effects of PLGA-PEI-si-m/hVDAC1-B versus PLGA-PEI-si-NT, with treatment starting at week 20 and terminating at week 38 following urethane treatment (Fig. 2A).
  • Experiment II: Comparison of the effects of PLGA-PEI-si-m/hVDAC1-B versus PLGA-PEI-si-NT, with treatment starting at week 21 or 23 and terminating at week 34 following urethane treatment (Fig. 3A).
  • Experiment III: Peptide treatment began at week 20 and ended at week 30 following urethane treatment new (Fig. 9A).
  • Experiment IV: Comparison of the effects of treatment with si-m/hVDAC1-B versus si-NT in an SCLC xenograft mice model (Fig. 7C, new).

The information regarding the test groups, number of mice, their age, the times of performing MRI analysis and the time of euthanasia were now added to section 2.6 of the Methods section.

The updated information, including details about the test groups, number of mice, their age, MRI analysis timings, and euthanasia, has been added to Section 6.2 of the Methods section and to the figure legends, and has been extended as described in our responses to Comments 1 and 3 below.

  1. Similarly, in section 2.8 (Xenograft mouse model), each group reflects 7 mice/group, but did all of them survive after treatment, or did a few die? Such information should be clearly reflected.

No mice died

  1. A chart or a tabular representation of the entire study design with a timeline for sections 2.6 and 2.8 should be given reflecting the no. of mice used, their initial and final weights, day of treatment, day of euthanization, etc., for clarity and better understanding.

As detailed in our response to Comment 1, we have added a chart for all experiments (new in Fig. 7 and 9) and updated Sections 2.6 and 2.8 of the Methods section and the figure legends to include information about the test groups, number of mice, their age, MRI analysis timings, and euthanasia.

  1. In Figure 1, part B, one of the blots of mouse cell line 2LL reflecting the VDAC1 expression level has some impurity as well because the blots reflect double lines. Check and, if necessary, change that blot image.

The antibodies used, Abcam 15895, recognizes both VDAC1 and VDAC3, thus the double bands represent these VDAC1 isoforms

  1. The B and D parts of Figure 3 need to be updated with a better-resolution image. Also, the B and C parts of Figure 6 need to be updated with a better-resolution image.

We have included the requested image with improved resolution/contrast. However, please note that the MRI or panoramic microscope used does not match the resolution of a confocal microscope. We chose this microscope to capture a broader area of the tumors or the whole lung (MRI).

  1. In Figure 7, part B, two of the shaded circles in black don't have any error bars. It needs to be rechecked and modified.

The bars are too small to see clearly. Therefore, for this reviewer's convenience, we present the figure below with the symbols significantly enlarged (can be seen in the attached PDF)

In Figure 8, part B, the significance shown with st westrrs has been shown vertically. Please maintain uniformity while using signs and symbols for significance, it should be horizontal everywhere.

This has been corrected now.

In Figure 9, part D, there is a misalignment in the significance shown, which needs to be corrected.

It has been corrected now.

Round 2

Reviewer 1 Report

Comments and Suggestions for Authors

The manuscript by Melnikov et al. presents compelling evidence that siRNA against VDAC1, when encapsulated in PLGA-PEI nanoparticles along with VDAC1-based peptides, can be administered intravenously and successfully target lung tumors. This targeted approach not only inhibits tumor growth but also reverses their oncogenic properties. This study demonstrated the effectiveness of the VDAC1-based peptide, Retro-Tf-D-LP4, in inhibiting tumor growth and reducing the expression of proteins associated with metabolism and cancer stem cells.

The authors have addressed all the previous comments. Thus, the manuscript can be accepted in its present form.

Author Response

We thank this reviewer for the valuable feedback, which has been instrumental in revising and improving our manuscript.